# PLASTIC LEARNING WITH DEEP FOURIER FEATURES

**Alex Lewandowski**[1,2]     **Dale Schuurmans**[1,2,3,4]     **Marlos C. Machado**[1,2,4]

[1]Deparment of Computing Science, University of Alberta, [2]Amii,
[3]Google DeepMind, [4]Canada CIFAR AI Chair

## ABSTRACT

Deep neural networks can struggle to learn continually in the face of non-stationarity, a phenomenon known as loss of plasticity. In this paper, we identify underlying principles that lead to plastic algorithms. We provide theoretical results showing that linear function approximation, as well as a special case of deep linear networks, do not suffer from loss of plasticity. We then propose *deep Fourier features*, which are the concatenation of a sine and cosine in every layer, and we show that this combination provides a dynamic balance between the trainability obtained through linearity and the effectiveness obtained through the nonlinearity of neural networks. Deep networks composed entirely of deep Fourier features are highly trainable and sustain their trainability over the course of learning. Our empirical results show that continual learning performance can be improved by replacing `ReLU` activations with deep Fourier features combined with regularization. These results hold for different continual learning scenarios (e.g., label noise, class incremental learning, pixel permutations) on all major supervised learning datasets used for continual learning research, such as CIFAR10, CIFAR100, and tiny-ImageNet.

## 1 INTRODUCTION

Continual learning is a problem setting that moves past some of the rigid assumptions found in supervised, semi-supervised, and unsupervised learning (Ring, 1994; Thrun, 1998). In particular, the continual learning setting involves learning from data sampled from a changing, non-stationary distribution rather than from a fixed distribution. A performant continual learning algorithm faces a trade-off due to its limited capacity: it should avoid forgetting what was previously learned while also being able to adapt to new incoming data, an ability known as plasticity (Parisi et al., 2019). Current approaches that use neural networks for continual learning are not yet capable of making this trade-off due to catastrophic forgetting (Kirkpatrick et al., 2017) and loss of plasticity (Dohare et al., 2021; Lyle et al., 2023; Dohare et al., 2024). The training of neural networks is in fact an active research area in the theory literature for supervised learning (Jacot et al., 2018; Yang et al., 2023; Kunin et al., 2024), which suggests there is much left to be understood in training neural networks continually. Compared to the relatively well-understood problem setting of supervised learning, even the formalization of the continual learning problem is an active research area (Kumar et al., 2023a; Abel et al., 2024; Liu et al., 2023). With these uncertainties surrounding current practice, we take a step back to better understand the inductive biases used to build algorithms for continual learning.

One fundamental capability expected from a continual learning algorithm is its sustained ability to update its predictions on new data. Recent work has identified the phenomenon of *loss of plasticity* in neural networks in which stochastic gradient-based training becomes less effective when faced with data from a changing, non-stationary distribution (Dohare et al., 2024). Several methods have been proposed to address the loss of plasticity in neural networks, with their success demonstrated empirically across both supervised and reinforcement learning (Ash and Adams, 2020; Lyle et al., 2022; 2023; Lee et al., 2024). Empirically, works have identified that the plasticity of neural networks is sensitive to different components of the training process, such as the activation function (Abbas et al., 2023). However, little is known about what is required for learning with sustained plasticity.

The goal of this paper is to identify a basic continual learning algorithm that does not lose plasticity in both theory and practice, rather than mitigating the loss of plasticity in existing neural network architectures. Our focus is on loss of plasticity rather than catastrophic forgetting, because plasticity is needed to sustain continual learning from new data. In particular, we investigate the effect of the nonlinearity of neural networks on the loss of plasticity. While loss of plasticity is a well-documented phenomenon in neural networks, previous empirical observations suggest that linear function

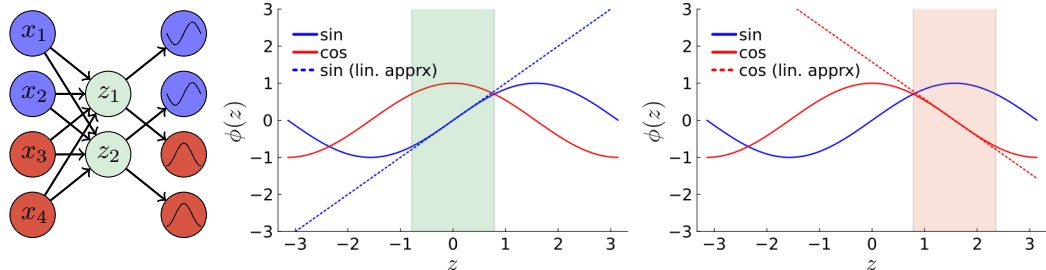

Figure 1: **A neural network with deep Fourier features in every layer approximately embeds a deep linear network.** A single layer using deep Fourier features linearly combines the inputs, $x$, to compute the pre-activations, $z$, and each pre-activation is mapped to both a `cos` unit and a `sin` unit (*Left*). For each pre-activation, either the `sin` unit (*Middle*) or the `cos` unit (*Right*) is well-approximated by a linear function.

approximation is capable of learning continually without suffering from loss of plasticity (Dohare et al., 2021; 2024). In this paper, we prove that linear function approximation does not suffer from loss of plasticity and can sustain their learning ability on a sequence of tasks. We then extend our analysis to a special case of deep linear networks, which provide an interesting intermediate case between deep nonlinear networks and linear function approximation. This is because deep linear networks are linear in representation but nonlinear in gradient dynamics (Saxe et al., 2014). We provide theoretical and empirical evidence that general deep linear networks also do not suffer from loss of plasticity. The plasticity of deep linear networks is surprising compared to the loss of plasticity of deep nonlinear networks. This finding suggests that loss of plasticity is not necessarily caused by the nonlinear learning dynamics, but a combination of nonlinear learning dynamics and nonlinear representations.

Given this seemingly natural advantage of linearity for continual learning, as well as its inherent limitation to learning only linear representations, we explore how nonlinear networks can better emulate the dynamics of deep linear networks to sustain plasticity. We hypothesize that, to effectively learn continually, the neural network must balance between introducing too much linearity and suffering from loss of deep representations and introducing too much nonlinearity and suffering from loss of plasticity. In fact, we show that previous work partially satisfies this hypothesis, such as the concatenated `ReLU` (Shang et al., 2016), `leaky-ReLU` activations (Xu et al., 2015), and residual connections (He et al., 2016), but they fail at striking this balance. Our results build on previous work that identified issues of unit saturation (Abbas et al., 2023) and unit linearization (Lyle et al., 2024) as issues in continually training neural networks with common activation functions. In particular, we generalize both to the problem of low *unit sign entropy*, which indicates a lack of diversity in the activations as measured by the entropy of the sign of the hidden units. We show that linear networks have high unit sign entropy, meaning that the sign of a hidden unit on different inputs is positive on approximately half the inputs. In contrast, deep nonlinear networks with most activation functions tend to have low unit sign entropy, which indicates saturation or linearization.

Periodic activation functions (Parascandolo et al., 2017), like the sinusoid function (`sin`), are a notable exception for having high unit sign entropy despite still suffering from loss of plasticity. Thus, in addition to unit sign entropy, we demonstrate that the network's activation function should be well-approximated by a linear function. We propose *deep Fourier features* as a means of approximating linearity dynamically, with every pre-activation being connected to two units, one of which will always be well-approximated by a linear function. In particular, deep Fourier features concatenate a sine and a cosine activation in each hidden layer. The resulting network is nonlinear while also approximately embedding a deep linear network using all of its parameters. Deep Fourier features differ from previous approaches that use Fourier features only in the input layer (Tancik et al., 2020; Li and Pathak, 2021; Yang et al., 2022) or that use fixed Fourier feature basis (Rahimi and Recht, 2007; Konidaris et al., 2011). We demonstrate that networks using these shallow Fourier features still exhibit a loss of plasticity. Only by using deep Fourier features in every layer is the network capable of sustaining and improving trainability in a continual learning setting. Using tiny-ImageNet (Le and Yang, 2015), CIFAR10, and CIFAR100 (Krizhevsky, 2009), we show that deep Fourier features can be used as a drop-in replacement for improving trainability in commonly used neural network architectures. Furthermore, the trainability of deep Fourier features enables training with a much larger regularization strength, leading to superior generalization performance.

## 2    PROBLEM SETTING

We define a deep network, $f_\theta$ with a a parameter set, $\theta = \{\mathbf{W}_l, \mathbf{b}_l\}_{l=1}^L$, as a sequence of layers, in which each layer applies a linear transformation followed by an element-wise activation function, $\phi$ in each hidden layer. The output of the network, $f_\theta(x) := h_L(x)$, is defined recursively by $h_l = [h_{l,1}, \dots, h_{l,w}] = [\phi(z_{l,1}), \dots, \phi(z_{l,w})] = \phi(z_l)$, and, $z_l = \mathbf{W}_l h_{l-1} + \mathbf{b}_l$ where $w$ is the width of the network, and $h_0 = x$. We refer to a particular element of the hidden layer's output $h_{l,i}$ as a unit. The deep network is a deep linear network when the activation function is the identity, $\phi(z) = z$. Linear function approximation is equivalent to a linear network with $L = 1$.

The problem setting that we consider is continual supervised learning, where the learner does not have information about the task boundaries. Instead, at each iteration the learner has access to a minibatch of observation-target pairs of size $M$, $\{x_i, y_i\}_{i=1}^M$. This minibatch is used to update the parameters $\theta$ of a neural network $f_\theta$ using a variant of stochastic gradient descent. The learning problem is continual because the distribution from which the data is sampled, $p(x, y)$, is changing. For simplicity, we assume this non-stationarity changes the distribution over the input-target pairs every $T$ iterations. The data is sampled from a single distribution for $T$ steps, and we refer to this particular temporary stationary problem as a task, $\tau$. The distribution over observations and targets that defines a task $\tau$ is denoted by $p_\tau$.

Loss of plasticity can refer to two related phenomena: loss of generalization (Ash and Adams, 2020; Dohare et al., 2024) or loss of trainability (Dohare et al., 2021; Lyle et al., 2023). We focus our theoretical analysis on the problem of loss of trainability, in which we evaluate the neural network at the end of each task using samples from the most recent task distribution, $p_\tau$, as is commonly done in previous work (Lyle et al., 2023). Loss of trainability refers to the problem where the neural network is unable to sustain its initial performance on the first task to later tasks. Specifically, we denote the optimisation objective by $J_\tau(\theta) = \mathbb{E}_{(x,y) \sim p_\tau}\left[\ell(f_\theta(x), y)\right]$, for some loss function $\ell$, and task-specific data distribution $p_\tau$. We use $t$ to denote the iteration count of the learning algorithm, and thus the current task number can be written as $\tau(t) = \lfloor t/T \rfloor$.

## 3    TRAINABILITY AND LINEARITY

In this section, we show that, unlike nonlinear networks, linear networks do not suffer from loss of trainability. That is, if the number of iterations in each task is sufficiently large, a linear network sustain trainability on every task in the sequence. We then show theoretically that a special case of deep linear networks also does not suffer from loss of trainability, and we empirically validate the theoretical findings in more general settings. These results provide a theoretical basis for previous work that uses a linear baseline in loss of plasticity experiments.

### 3.1    TRAINABILITY OF LINEAR FUNCTION APPROXIMATION

We first prove that loss of trainability does not occur with linear function approximation, $f_\theta(x) = \mathbf{W}_l x + \mathbf{b}_l$. We prove this by showing that linear function approximators can sustain learning on a sequence of tasks, with a large enough number of iterations per task. In particular, the performance of the solution found on the $\tau$-th task can be upper bounded on a quantity that is independent of the solution found on the first $\tau - 1$ tasks. Linear function approximation avoids loss of trainability because the optimisation problem on each task is convex (Agrawal et al., 2021; Boyd and Vandenberghe, 2004), with a unique global optimum, $\theta_\tau^\star$. We now state the theorem, which we prove in Appendix B

**Theorem 1.** *Let* $\theta^{(\tau T)}$ *denote the linear weights learned at the end of the $\tau$-th task, with the corresponding unique global minimum for task $\tau$ being denoted by $\theta_\tau^\star$. Assuming the objective function is $\mu$-strongly convex, the suboptimality gap for gradient descent on the $\tau$-th task is*

$$J_\tau(\theta^{(\tau T)}) - J_\tau(\theta_\tau^\star) < \frac{2D(1 - \alpha\mu)^T}{\alpha T(1 - (1 - \alpha\mu)^T)},$$

*where each task lasts for $T$ iteration, $D$ is the assumed bound on the parameters at the global minimum for every task, and $\alpha$ is the step-size.*

Intuitively, this theorem states that if the problem is bounded and effectively strongly convex due to a finite number of iterations, then the optimisation dynamics are well-behaved for every task in the bounded set. In particular, this means that the error on each task can be upper bounded by a quantity independent of the initialization found on previous tasks. Thus, given enough iterations, linear function approximation can learn continually without loss of trainability.

## 3.2 TRAINABILITY OF DEEP LINEAR NETWORKS

We now provide evidence that, similar to linear function approximation, deep linear networks also do not suffer from loss of trainability. Unlike deep nonlinear networks, deep linear networks use linear activation functions in their hidden layers (Bernacchia et al., 2018; Ziyin et al., 2022). This means that a deep linear network can only represent linear functions. At the same time, its gradient update dynamics are nonlinear and non-convex, similar to deep nonlinear neural networks (Saxe et al., 2014). Our central claim here is that *deep linear networks under gradient descent dynamics avoid parameter configurations that would lead to loss of trainability*.

To simplify notation, without loss of generality, we combine the weights and biases into a single parameter for each layer in the deep linear network , $\theta = \{\theta_1, \ldots, \theta_L\}$, and $f_\theta(x) = \theta_L \theta_{L-1} \cdots \theta_1 x$. We denote the product of weight matrices, or simply product matrix, as $\bar{\theta} = \theta_L \theta_{L-1} \cdots \theta_1$, which allows us to write the deep linear network in terms of the product matrix: $f_\theta(x) = \bar{\theta} x$. The problem setup we use for the deep linear analysis follows previous work (Huh, 2020), and we provide additional technical details for optimisation dynamics of deep linear networks in Appendix A.3.

We now provide evidence to suggest that, despite deep linear networks being nonlinear in their gradient dynamics, they do not suffer from loss of trainability. We prove this for a special case of deep diagonal linear networks, and provide empirical evidence to support this claim in general deep linear networks.

**Theorem 2.** *Let $f_\theta(x) = \theta_L \theta_{L-1} \cdots \theta_1 x$ be a deep diagonal linear network where $\theta_l = Diag(\theta_{l,1}, \ldots, \theta_{l,d})$. Then, a deep diagonal linear network converges on a sequence of tasks under the same conditions for convergence in a single task (i.e., the conditions in Arora et al., 2019).*

Theorem 2 states that a deep diagonal linear network, a special case of general deep linear networks, can converge to a solution on each task within a sequence of tasks. The proof, provided in Appendix B, shows that the minimum singular value of the product matrix stays greater than zero, $\sigma_{min}(\bar{\theta}) > 0$. Hence, deep diagonal linear networks do not suffer from loss of trainability. This result provides further evidence suggesting that linearity might be an effective inductive bias for learning continually.

While the analysis considers a special case of deep linear networks, namely deep diagonal networks, we note that this is a common setting for the analysis of deep linear networks more generally (Nacson et al., 2022; Even et al., 2023). In particular, the analysis is motivated by the fact that, under certain conditions, the evolution of the deep linear network parameters can be analyzed through the independent singular mode dynamics (Braun et al., 2022), which simplifies the analysis of deep linear networks to deep diagonal linear networks.

## 3.3 EMPIRICAL EVIDENCE FOR TRAINABILITY OF GENERAL DEEP LINEAR NETWORKS

In the previous section, we proved that a special case of deep linear networks do not suffer from loss of trainability. We now provide additional empirical evidence that general deep linear networks do not suffer from loss of trainability. To do so, we use a linearly separable subset of the MNIST dataset (LeCun et al., 1998), in which the labels of each image are randomized every 100 epochs. For this experiment, the data is linearly separable so that even a linear baseline can fit the data if given enough iterations. While MNIST is a simple classification problem, memorizing random labels highlights the difficulties associated with maintaining trainability (see Lyle et al., 2023; Kumar et al., 2023b). We emphasize that the goal here is merely to validate that linear networks remain trainable in continual learning. We also provide results with traditional nonlinear neural networks on the same problem, showing that they suffer from loss of trainability in this simple problem. Later in Section 5, we extend our investigation of loss of trainability to larger-scale benchmarks.

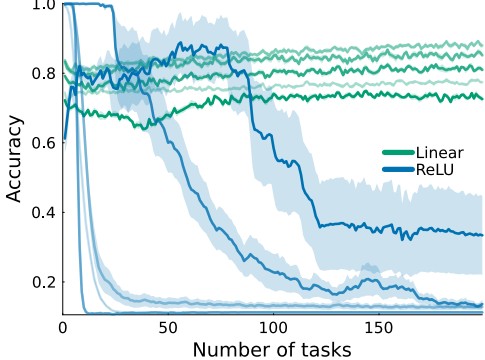

Figure 2: **Trainability on a linearly separable task.** The higher opacity corresponds to deeper networks, ranging from {1, 2, 4, 8, 16}. Deep linear networks sustain trainability on new tasks, with some additional depth improving trainability. Nonlinear networks, using ReLU, suffer from loss of trainability at any depth even on this simple sequence of linearly separable problems.

In Figure 2, we see that deep linear networks ranging from a depth of 1 to 16 can sustain trainability. Using a multi-layer perceptron with `ReLU` activations, deep nonlinear networks quickly reach a much higher accuracy on the first few tasks. However, due to loss of trainability, deep nonlinear networks of any depth eventually perform worse than the corresponding deep linear network. With additional epochs, the linear networks could achieve perfect accuracy on this task because it is linear separable. The number of epochs is comparatively low to showcase that, with some additional layers, a deep linear network is able to improve its trainability as new tasks are encountered.

## 4    COMBINING LINEARITY AND NONLINEARITY

In the previous section, we provided empirical and theoretical evidence that linearity provides an effective inductive bias for learning continually by avoiding loss of trainability. However, linear methods are generally not as performant as deep nonlinear networks, meaning that their sustained performance can be inadequate on complex tasks. Even deep linear networks have only linear representational power, despite their nonlinear gradient dynamics. We now seek to answer the following question:

> *How can the sustained trainability of linear methods be combined with*
> *the expressive power of learned nonlinear representations?*

To answer this question, we first seek to better understand the effects of replacing linear activation functions with nonlinear ones in deep networks for continual learning. We observe that deep linear networks have diversity in their hidden units, which can be induced in nonlinear activation functions by adding linearity through a weighted linear component, an idea we refer to as $\alpha$-*linearization*. To dynamically balance linearity and nonlinearity, we propose to use *deep Fourier features* for every layer in a network. We prove that such a network approximately embeds a deep linear network, a property we refer to as *adaptive linearity*. We demonstrate that this adaptively-linear network is plastic, maintaining trainability even on non-linearly-separable problems.

### 4.1    ADDING LINEARITY TO NONLINEAR ACTIVATION FUNCTIONS

Deep nonlinear networks can learn expressive representations because of their nonlinear activation functions, but these nonlinearities can also lead to issues with trainability. Although several components of common network architectures incorporate linearity, the way in which linearity is used does not avoid loss of trainability. One example is the piecewise linearity of the ReLU activation function (Shang et al., 2016), $\text{ReLU}(x) = \max(0, x)$, which is said to be *saturated* if $\text{ReLU}(x) = 0$ for most inputs $x$, preventing gradient propagation. While saturation is generally not a problem for learning on a single distribution, it has been noted as problematic in learning from changing distributions, for example, in reinforcement learning (Abbas et al., 2023).

A potential solution to saturation is to use a non-saturating activation function. Two noteworthy examples of non-saturating activation functions include a periodic activation like $\sin(x)$ (Parascandolo et al., 2017) and $\text{leaky-ReLU}_\alpha(x) = \alpha x + (1 - \alpha)\text{ReLU}(x)$ (Xu et al., 2015), both of which are zero on a set of measure zero. Surprisingly, using `leaky-ReLU` leads to a related issue, "unit linearization" (Lyle et al., 2024), in which the activation is only positive (or negative) for most inputs $x$. Unlike saturated units, linearized units can provide non-zero gradients but render that unit effectively linear, limiting the expressive power of the learned representation. While unit linearization seems to suggest that loss of trainability can occur due to linearity, it is important to note that a "linearized unit" is not the same as a linear unit. This is because a linearized unit provides mostly positive (or negative) outputs, whereas a linear unit can output both positive and negative values.

We generalize the idea behind unit saturation and unit linearization to *unit sign entropy*, which is applicable to activation functions beyond saturating and piecewise linear functions, such as periodic activation functions. Intuitively, it measures the diversity of the activations of a hidden layer.

**Definition 1** (Unit Sign Entropy). *The entropy, $\mathbb{H}$, of the unit's sign, $sgn(h(x))$, on a distribution of inputs to the network, $p(x)$, is given by $\mathbb{H}\left(sgn(h(x))\right) = \mathbb{E}_{p(x)}\left[sgn(h(x))\right]$.*

The maximum value of unit sign entropy is 1, which occurs when the unit is positive on half the inputs. Conversely, a low sign entropy is associated with the aforementioned issues of saturation and linearization. For example, a low sign entropy for a deep network using `ReLU` activations means that the unit is almost always positive ($P\left(\text{sgn}(h(x)) = 1\right) = 1$, meaning it is linearized) or negative ($P\left(\text{sgn}(h(x)) = 1\right) = 0$, meaning it is saturated).

With unit sign entropy, we investigate how the leak parameter for the `leaky-ReLU` activation function influences training as pure linearity ($\alpha = 1$) is traded-off for pure nonlinearity ($\alpha = 0$). The idea of mixing a linearity and nonlinearity can also be generalized to an arbitrary activation function, which we refer to as the $\alpha$-linearization of an activation function.

**Definition 2** ($\alpha$-linearization). *The $\alpha$-linearization of an activation function $\phi$, is denoted by $\phi_\alpha(x) = \alpha x + (1 - \alpha)\phi(x)$.*

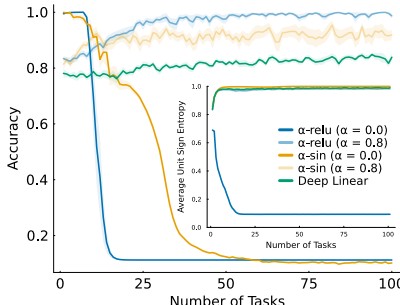

A natural hypothesis is that, as $\alpha$ increases from 0 to 1, and the network becomes more linear, loss of trainability is mitigated. We emphasize that the $\alpha$-linearization is primarily to gain insights from empirical investigation and it is not a solution to loss of trainability. This is because any benefits of $\alpha$-linearization depend on tuning $\alpha$, and even optimal tuning can lead to overly linear representations and slow training compared to nonlinear networks.

**Empirical Evidence for $\alpha$-linear Plasticity**   To understand the trainability issues introduced by nonlinearity, we present a case-study using `sin` and `ReLU` with different values of the linearization parameter, $\alpha$. The same experiment setup is used from Section 3.3. Referring to the results in Figure 3, we see that both `ReLU` and `sin` activation functions are able to sustain trainability for larger values of $\alpha$. This verifies the hypothesis: a larger $\alpha$ pro-

Figure 3: **Trainability on a linearly separable task with $\alpha$-linearization** Darker opacity lines correspond to higher values of $\alpha$. Unit sign entropy increases as $\alpha$ increases (*inset*), leading to sustained trainability for $\alpha$-relu.

vides more linearity to the network, allowing it to sustain trainability. Despite sustaining trainability, a larger $\alpha$ can lead to overly linear representations, evidenced by worse performance and slower training speed on the first few tasks compared to nonlinear networks ($\alpha = 0$). For $\alpha$-`ReLU`, we also verify the hypothesis that the unit sign entropy increases for larger values of $\alpha$ (inset plot). The fact that the periodic `sin` activation function has a high unit sign entropy despite losing trainability is particularly interesting, which we explore in the next section.

## 4.2   Adaptive-linearity by Concatenating Sinusoid Activation Functions

Using the insight that linearity promotes unit sign entropy, we explore an alternative approach to sustain trainability. In particular, we found that linearity can sustain trainability but requires tuning $\alpha$, and even optimal tuning can lead to slow learning from overly linear representations. Our approach is motivated by concatenated `ReLU` activations (Shang et al., 2016; Abbas et al., 2023), $\text{CReLU}(z) = [\text{ReLU}(z), \text{ReLU}(-z)]$, which avoids the problems from saturated units, but does not avoid the problem of low unit sign entropy. In particular, we propose using a pair of activations functions such that one activation function is always approximately linear, with a bounded error.

One way to dynamically balance the linearities and nonlinearities of a network is using periodic activation functions. This is because, due to their periodicity, the properties of the activation function can re-occur as the magnitude of the preactivations grows rather than staying constant, linear, or saturating. But, as we saw in Figure 3, a single periodic activation function like `sin` is not enough. Instead, we propose to use deep Fourier features, meaning that every layer in the network uses Fourier features. This is a notable departure from previous work which considers only shallow Fourier features in the first layer (Rahimi and Recht, 2007; Tancik et al., 2020). In particular, each unit is a concatenation of a sinusoid basis of two elements, $\text{Fourier}(z) = [\sin(z), \cos(z)]$. Each pre-activation is mapped to both $\sin(z)$ and $\cos(z)$, which requires that a layer with deep Fourier features have half the output width to accomodate the concatenation.[1]

The advantage of this approach is that a network with deep Fourier features maintains approximate linearity in all of its parameters. Moreover, deep Fourier features are closed under differentation, meaning that the activations and their gradients provide a basis for representing periodic functions .

**Proposition 1.** *For any $z$, there exists a linear function, $L_z(x) = a(z)x + b(z)$, such that either:* $|\sin(x) - L_z(x)| \leq c$, *or* $|\cos(x) - L_z(x)| \leq c$, *for* $c = \sqrt{2}\pi^2/2^8$ *and all* $x \in [z - \pi/4, z + \pi/4]$.

---

[1]For a fixed width, a network with deep Fourier features has approximately half the number of parameters.

An intuitive description of this is provided in Figure 1. The advantage of using two sinusoids over just a single sinusoid is that whenever $\cos(z)$ is near a critical point, $d/dz \cos(z) \approx 0$, we have that $\sin(z) \approx z$, meaning that $d/dz \sin(z) \approx 1$ (and vice-versa). The argument follows from an analysis of the Taylor series remainder, showing that the Taylor series of half the units in a deep Fourier layer can be approximated by a linear function, with a small error of $c = \sqrt{2}\pi^2/2^8 \approx 0.05$. While we found that two sinusoids is sufficient, the approximation error can be further improved by concatenating additional sinusoids, at the expense of reducing the effective width of the layer.

Because each pre-activation is connected to a unit that is approximately linear, we can conclude that a deep network comprised of deep Fourier features approximately embeds a deep linear network.

**Corollary 1.** *A network parameterized by $\theta$, with deep Fourier features, approximately embeds a deep linear network parameterized by $\theta$ with a bounded error.*

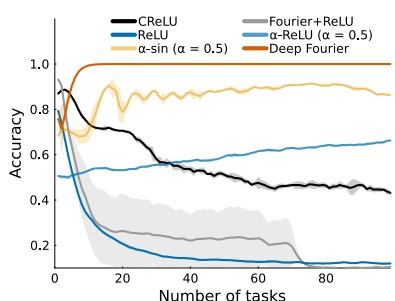

Notice that piecewise linear activations also embed a deep linear network, but these embedded deep linear networks do not use the same parameter set. For example, the deep linear network embedded by a `ReLU` network does not depend on any of the parameters used to compute a `ReLU` unit that is zero. Although the `leaky-ReLU` function involves every parameter, the deep linear network vanishes because the leak parameter is small, $\alpha < 1$, and hence the embedded deep linear network is multiplied by a small constant, $\alpha^{-L}$, where $L$ is the depth of the network.

**Empirical Evidence for Nonlinear Plasticity** We now consider a similar experimental setup from Sections 3.3 and 4.1, except we make the problem non linearly-separable by considering random label assignments on the entire dataset. Each task is more difficult because

Figure 4: **Trainability on a non linearly-separable task.** Deep Fourier features improve and sustain their trainability when other networks cannot.

it involves memorizing more labels, and the effect of the non-stationarity is also stronger due to randomization of more datapoints. As a result, the deep linear network can no longer fit a single task well. Referring to Figure 4, the $\alpha$-linear activation functions can sustain and even improve their trainability, albeit very slowly. See also unit sign entropy in Figure 9, Appendix D.1. In contrast, using deep Fourier features within the network enables the network to easily memorize all the labels for 100 tasks. Deep Fourier features surpass the trainability of the other nonlinear baselines at initialization, `CReLU` and shallow Fourier features followed by `ReLU`. This is surprising, because deep nonlinear networks at initialization are often a gold-standard for trainability.

## 5 EXPERIMENTS

Our experiments demonstrate the benefits of the adaptive linearity provided by deep Fourier features. While trainability was the primary focus behind our theoretical results and empirical case studies, we show that these findings generalize to other problems in continual learning. In particular, we demonstrate that networks composed of deep Fourier features are capable of being strongly regularized leading to improved generalization performance on diminishing levels of label noise, and in class-incremental learning. The main results we present are on all of the major continual supervised learning settings considered in the plasticity literature. They build on the standard ResNet-18 architecture, widely used in practice (He et al., 2016).

**Datasets and Non-stationarities** Our experiments use the common image classification datasets for continual learning, namely tiny-ImageNet (Le and Yang, 2015), CIFAR10, and CIFAR100 (Krizhevsky, 2009). We augment these datasets with commonly used non-stationarities to create continual learning problems, with the non-stationarity creating a sequence of tasks from the dataset. Specifically, we follow recent work that introduced the diminishing label noise problem (Lee et al., 2024), which is inspired by the warm-starting problem: We start with half the data being corrupted by label noise and reduce the noise to clean labels over 10 tasks. Additionally, for the datasets with a larger number of classes, tiny-ImageNet and CIFAR100, we also consider the class-incremental setting: the first task involves only five classes, and five new classes are added to the existing pool of classes at the beginning of each task (Van de Ven et al., 2022). Other results and more details on datasets and non-stationarities considered can be found in Appendix C.

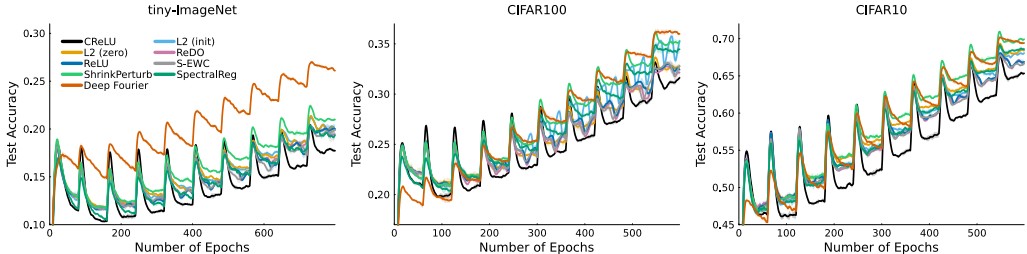

Figure 5: **Training a ResNet-18 continually with diminishing label noise.** Deep Fourier features are particularly performant on complex tasks like tiny-ImageNet. Despite networks with deep Fourier features having approximately half the number of parameters, they surpass the baselines in CIFAR100 and are on-par with spectral regularization on CIFAR10.

**Architecture and Baselines** We compare a ResNet-18 using only deep Fourier features against a standard ResNet-18 with `ReLU` activations. The network with deep Fourier features has fewer parameters because it uses a concatenation of two different activation functions, halving the effective width compared to the network with `ReLU` activations. This provides an advantage to the nonlinear baseline. We also include *all* prominent baselines that have previously been proposed to mitigate loss of plasticity in the field: L2 regularization towards zero, L2 regularization towards the initialization (Kumar et al., 2023b), spectral regularization (Lewandowski et al., 2024), Concatenated ReLU (Shang et al., 2016; Abbas et al., 2023), Dormant Neuron Recycling (ReDO, Sokar et al., 2023), Shrink and Perturb (Ash and Adams, 2020), and Streaming Elastic Weight Consolidation (S-EWC, Kirkpatrick et al., 2017; Elsayed and Mahmood, 2024).

## 5.1 MAIN RESULTS

Our results demonstrate that deep Fourier features, combined with regularization, are effective at continual learning. In these set of experiments, we consider the problem of sustaining test accuracy on a sequence of tasks. In addition to requiring trainability, methods must also sustain generalization.

**Diminishing Label Noise** In Figure 5, we can clearly see the benefits of deep Fourier features in the diminishing label noise setting. At the end of training on ten tasks with diminishing levels of label noise, the network with deep Fourier features was always among the methods with the highest test accuracy on the the uncorrupted test set. On the first of ten tasks, deep Fourier features could occasionally overfit to the corrupted labels leading to initially low test accuracy. However, as the label noise diminished on future tasks, the network with deep Fourier features was able to continue to learn to correct its previous poorly-generalizing predictions. In contrast, the improvements achieved by the other methods that we considered was oftentimes marginal compared to the baseline `ReLU` network. Two exceptions are: (i) networks with `CReLU` activations, which underperformed relative to the baseline network, and (ii) Shrink and Perturb, which was the best-performing baseline method for diminishing label noise. Interestingly, the performance benefit of deep Fourier features is most prominent on more complex datasets, like tiny-ImageNet. In Appendix D.2, we provide an ablation of the architecture, where we use a Wide Residual Network (Zagoruyko and Komodakis, 2016) and vary the width scale.

**Class-Incremental Learning** Deep Fourier features are also effective in the class-incremental setting, where later tasks involve training on a larger subset of the classes, following the experiment described in (Dohare et al., 2024). The network is evaluated at the end of each task on the entire test set. As the network is trained on later tasks, its test set performance increases

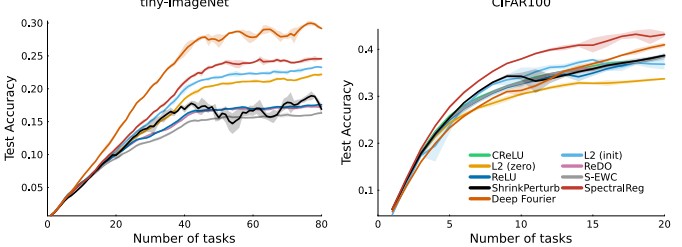

Figure 6: **Class incremental learning results on tiny-Imagenet (*Left*) and CIFAR-100 (*Right*).** On both datasets, deep Fourier features substantially improve over most baselines.

because it has access to a larger subset of the training data. In Figure 6, we see that Deep Fourier features largely outperform the baselines in this setting, particularly on tiny-ImageNet in which the first forty tasks involve training on a growing subset of the dataset and the last forty "tasks" involve training to convergence on the full dataset. We use quotation marks to characterize the last forty tasks because they are, in fact, a single task, as the data distribution stops changing after the first forty tasks. We call them "tasks" because of the number of iterations in which they are trained. Not only are deep Fourier features quicker to learn on earlier continual learning tasks, but they are also able to improve their generalization performance by subsequently training on the full dataset. On CIFAR100, the difference between methods is not as prominent, but we can see that deep Fourier features are still among the top-performing methods. The large performance difference on tiny-ImageNet can be attributed to the fact that it is a harder problem compared to CIFAR10 and CIFAR100, with higher resolution images, more classes and more datapoints.

## 5.2 SENSITIVITY ANALYSIS

In the previous sections, we found that deep Fourier features used in combination with spectral regularization leads to strong generalization performance. However, the theoretical analysis and case-studies that we presented earlier concerned trainability. We now present a sensitivity result to understand the relationship between trainability and generalization. Using a ResNet-18 with different activation functions, we varied the regularization strength between no regularization (left) and high regularization (right). In Figure 5.2, we can see that deep Fourier features indeed have a high degree of trainability, sustaining higher trainability at every level of regularization strength. However, without any regularization, deep Fourier features have a tendency to overfit. Over-fitting is a known issue for shallow

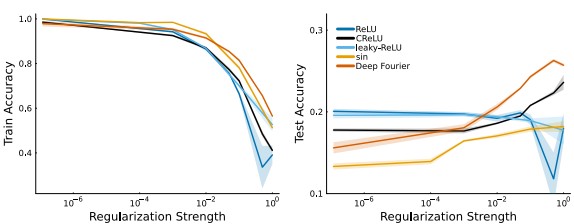

Figure 7: **Sensitivity analysis on tiny-ImageNet.** Networks with deep Fourier features are highly trainable, but have a tendency to overfit without regularization, leading to high training accuracy but low test accuracy. Due to deep Fourier features being highly trainable, they are able to train with much higher regularization strengths leading to ultimately better generalization.

Fourier features (e.g., when using Fourier features only for the input layer, Mavor-Parker et al., 2024), and this can be attributed to their spectral bias of learning high-frequency features (Tancik et al., 2020). However, deep Fourier features are able to use their high trainability to learn effectively even when highly regularized. Thus, while high trainability does not always lead to high generalization, the trainability provided by deep Fourier features can be used in combination with regularization to improve continual learning performance. Hyperparameter sensitivity results are presented on other datasets in Appendix D.6. We also provide an in-depth sensitivity study on smaller-scale MLPs in Appendix D.8.

## 6 CONCLUSION

In this paper, we proved that linear function approximation and a special case of deep linearity are effective inductive biases for learning continually without loss of trainability. This surprising finding for deep linear networks suggests that nonlinearity of representations, rather than nonlinearity of gradient dynamics, contributes to loss of plasticity. We then investigated the issues that arise from using nonlinear activation functions, namely the problem of low unit sign entropy, which indicates a lack of diversity in the activations as measured by the entropy of the sign of the hidden units. Motivated by the effectiveness of linearity in sustaining trainability, we proposed deep Fourier features to approximately embed a deep linear network inside a deep nonlinear network. We found that deep Fourier features dynamically balance the trainability afforded by linearity and the effectiveness of nonlinearity, thus providing an effective inductive bias for learning continually. Experimentally, we demonstrated that networks with deep Fourier features provided benefits for continual learning across every dataset we considered. We found that networks with deep Fourier features were effective plastic learners because their trainability enabled training with a much larger regularization strength, leading to superior generalization performance.

ACKNOWLEDGMENTS

The research is supported in part by the Natural Sciences and Engineering Research Council of Canada (NSERC), the Canada CIFAR AI Chair Program, Alberta Innovates, and the Digital Research Alliance of Canada.

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

# A    ADDITIONAL DETAILS

## A.1    ASSUMPTIONS FOR TRAINABIILTY OF LINEAR FUNCTION APPROXIMATION

We assume that the parameters at the global optimum for every task are bounded: $\|\theta_\tau\|_2 < D$. This is true for regression problems if the observations and targets are bounded. In classification tasks, the global optimum can be at infinity because activation functions such as the sigmoid and the softmax are maximized at infinity. In this case, we constrain the parameter set, $\{\theta : \|\theta\|_2 < D\}$, and project the optimum onto this set.

In addition to convexity, we assume that the objective function is $\mu$-strongly convex, $\nabla_\theta^2 J_\tau(\theta) \succ \mu\mathbf{I}$, where $\nabla_\theta^2 J_\tau(\theta)$ denotes the Hessian. Note that neither squared nor cross-entropy loss are $\mu$-strongly in general. However, this assumption is satisfied in continual learning problems with a finite number of iterations. For regression, denote the observations for task $\tau$ as $X_\tau \in mathbbR^{d \times N}$ where $N$ is the sample size. Then the Hessian is the outer products of the data matrix, $\nabla_\theta^2 J_\tau^{reg}(\theta) = X_\tau X_\tau^\top \in \mathbb{R}^{d \times d}$. Thus, the squared loss is strongly-convex if the data is full rank. This is satisfied in high dimensional image classification problems, which is what we consider.

For classification, the Hessian involves an additional diagonal matrix of the predictions for each datapoint,
$$\nabla_\theta^2 J_\tau^{class}(\theta) = X_\tau X_\tau^\top \in \mathbb{R}^{d \times d},$$
where $D = \mathrm{Diag}(p_1, \dots, p_N)$ and $p_i = 2\sigma(f_\theta(x_i))(1 - \sigma(f_\theta(x_i)))$. If the prediction becomes sufficiently confident, $\sigma(f_\theta(x_i)) = 1$, then there can be rank deficiency in the Hessian. However, because each task is only budgeted a finite number of iterations this bounds the predictions away from 1.

## A.2    RELATED WORK REGARDING TRAINABILITY OF DEEP LINEAR NETWORKS

Some authors have suggested deep linear networks suffer from a related issue, namely that critical learning periods also occur for deep linear networks (Kleinman et al., 2024). Unlike the focus on loss of trainability in this work where the entire network is trained, these critical learning periods are due to winner-take-all dynamics due to manufactured defects in one half of the linear network, for which the other half compensates.

Finally, we note that some previous work have found that gradient dynamics have a low rank bias for deep linear networks (Chou et al., 2024). One important assumption that these works make is that the neural network weights are initialized identically across layers, $\theta_j = \alpha\theta_1$. Our analysis assumes that the initialization uses small random values, such as those used in practice with common neural network initialization schemes (Glorot and Bengio, 2010; He et al., 2015).

## A.3    DETAILS FOR DEEP LINEAR SETUP

The gradient of the loss function with respect to the parameters of a deep linear network can be written in terms of the gradient with respect to the product matrix $\bar{\theta}$ (Bah et al., 2022):
$$\nabla_{\theta_j} J(\theta) = \theta_{j+1}^\top \theta_{j+2}^\top \cdots \theta_L^\top \nabla_{\bar{\theta}} J(\bar{\theta}) \theta_1^\top \theta_2^\top \cdots \theta_{j-1}^\top, \tag{1}$$

where the term $\nabla_{\bar{\theta}} J(\bar{\theta})$ is the gradient of the loss with respect to the product matrix, treating it as if it was linear function approximation. The gradient is nonlinear because of the coupling between the gradient of the parameter at one layer and the value of the parameters of the other layers. Nevertheless, the gradient dynamics of the individual parameters can be combined to yield the dynamics of the product matrix (Arora et al., 2018),
$$\bar{\nabla}_\theta J(\theta) = P_{\bar{\theta}} \nabla_{\bar{\theta}} J(\bar{\theta}). \tag{2}$$

The dynamics involve a preconditioner, $P_{\bar{\theta}}$, that accelerates optimisation (Arora et al., 2018), which we empirically demonstrate in Section 3.3. On the left-hand side of the equation, we use $\bar{\nabla}_\theta J(\theta)$ to denote the combined dynamics of the gradients for each layer on the dynamics of the product matrix.[2] This means that the effective gradient dynamics of the deep network is related to the dynamics of

---

[2]Note we use $\bar{\nabla}$ because $\bar{\nabla} J(\theta)$ is not a gradient for any function of $\bar{\theta}$; see discussion by Arora et al. (2018).

linear function approximation with a precondition. While the dynamics are nonlinear and non-convex, the overall dynamics are remarkably similar to that of linear function approximation, which is convex.

To simplify notation, without loss of generality, we consider a deep linear network without the bias terms, $\theta = \{\theta_1, \dots, \theta_L\}$, and $f_\theta(x) = \theta_L \theta_{L-1} \cdots \theta_1 x$. We denote the product of weight matrices, or simply product matrix, as $\bar{\theta} = \theta_L \theta_{L-1} \cdots \theta_1$, which allows us to write the deep linear network in terms of the product matrix: $f_\theta(x) = \bar{\theta}x$. The problem setup we use for the deep linear analysis follows previous work (Huh, 2020), and we provide additional details in Appendix A.3. We consider the squared error, $J_\tau(\theta) = \mathbb{E}_{(x,y) \sim p_\tau} \left[ \|y - \bar{\theta}x\| \right]_2^2$. and we assume that the observations are whitened to simplify the analysis,

$\Sigma_x = \mathbb{E}\left[ xx^\top \right] = \mathbf{I}$, focusing on the case where the targets $y$ are changing during continual learning. Then we can write the squared error as

$$J(\theta) = \text{Tr}\left[ \Delta_\tau \Delta_\tau^\top \right],$$

where $\Delta_\tau = \theta_\tau^\star - \bar{\theta}$ is the distance to the optimal linear predictor, $\theta_\tau^\star = \Sigma_{yx,\tau} = \mathbb{E}_{x,y \sim p_\tau}[yx^\top]\Sigma_x$.

The convergence of gradient descent for general deep linear networks requires an assumption on the deficiency margin, which is used to ensure that the solution found by a deep linear network, in terms of the product matrix, is full rank (Arora et al., 2019). That is, the deep linear network converges if the minimum singular value of the product matrix stays positive, $\sigma_{min}(\bar{\theta}) > 0$.

We now show that a diagonal linear network maintains a positive minimum singular value under continual learning. This is a simplified setting for analysis, where we assume that the weight matrices are diagonal and thus the input, hidden, and output dimension are all equal. Let $f_\theta(x)$ be a diagonal linear network, defined by a set of diagonal weight matrices, $\theta_l = \text{Diag}(\theta_{l,1}, \dots, \theta_{l,d})$. The output of the diagonal linear network is the product of the diagonal matrices, $f_\theta(x) = \theta_L \theta_{L-1} \dots \theta_1 x$. Then the product matrix is also a diagonal matrix, whose diagonals are the products of the parameters of each layer, $\bar{\theta} = Diag(\prod_{l=1}^L \theta_{l,1}, \dots, \prod_{l=1}^L \theta_{l,d}) := Diag(\bar{\theta}_1, \dots, \bar{\theta}_d)$. The minimum singular value of a diagonal matrix is the minimum of its absolute values, $\sigma_{min}(\bar{\theta}) = \min_i |\bar{\theta}_i|$. Thus, we must show that the minimum absolute value of the product matrix is never zero.

**Lemma 1.** *Consider a deep diagonal linear network, $f_\theta(x) = \theta_L \theta_{L-1} \dots \theta_1 x$ and $\theta_l = Diag(\theta_{l,1}, \dots, \theta_{l,d})$. Then, under gradient descent dynamics, $\theta_{l,i}^{(t)} = \theta_{l',i}^{(t)}$ iff $\theta_{l,i}^{(0)} = \theta_{l',i}^{(0)}$ for $l' \neq l$.*

The proof of this proposition, and the next, can be found in Appendix B. This first proposition states that two parameters that are initialized to different values, such as by a random initialization, will never have the same value under gradient descent. Conversely, if the parameters are initialized identically, then they will stay the same value under gradient descent. This means that, in particular, two parameters will never be simultaneously zero.

**Lemma 2.** *Denote a deep diagonal linear network as $f_\theta(x) = Diag(\bar{\theta}_1, \dots, \bar{\theta}_d)x$ where $\bar{\theta}_i = \prod_{l=1}^L \theta_{l,i}$. Then, under gradient descent dynamics, $\bar{\theta}_i^{(t)} = \bar{\theta}_i^{(t+1)} = 0$ iff two (or more) components are zero, $\theta_{l,i}^{(t)} = \theta_{l',i}^{(t)} = 0$, for $l' \neq l$.*

While the analysis considers a special case of deep linear networks, namely deep diagonal networks, we note that this is a common setting for the analysis of deep linear networks more generally (Nacson et al., 2022; Even et al., 2023). In particular, the analysis is motivated by the fact that, under certain conditions, the evolution of the deep linear network parameters can be analyzed through the independent singular mode dynamics (Saxe et al., 2014), which simplify the analysis of deep linear networks to deep diagonal linear networks. The target function being learned, $y^\star(x) = \theta^\star x$, is represented in terms of the singular-value decomposition, $\theta^\star = U^\star S^\star V^\star = \sum_{j=1}^r s_i u_i v_i^\top$. We also assume that the neural network has a fixed hidden dimension, so that $\theta_1 \in \mathbb{R}^{d \times d_{in}}, \theta_L \in \mathbb{R}^{d_{out} \times d}, \theta_{1 < l < L} \in \mathbb{R}^{d \times d}$; and we apply the singular value decomposition to the function approximator's parameters, $\theta_l = U_l S_l V_l \in \mathbb{R}^{d_{out} \times d_h}$. To simplify the product of weight matrices, we assume $V_{i+1} = U_i$, $V_1 = V^\star$, and $U_L = U^\star$. The simplifying result is that the squared error loss can be expressed entirely in terms of the singular values, $\|y^\star x - \prod_{i=L}^1 \theta_i x\|^2 \propto \|S^\star - \prod_{i=L}^1 S_l\|^2$, which is equivalent to our analysis of the deep diagonal network, as the matrix of singular values is a diagonal matrix. These decoupled learning dynamics are closely approximated by networks with small random weights and they persist under gradient flows (Huh, 2020).

A.4   PSEUDOCODE FOR DEEP FOURIER FEATURE LAYER

---

**Algorithm 1** Deep Fourier Feature Layer

---

 1: **function** DEEPFOURIERFEATURES($x, W, b$)
 2:      $z \leftarrow \mathbf{W}x + \mathbf{b}$                                    ▷ Calculate pre-activation
 3:      $a_1 \leftarrow \sin(z)$                                    ▷ Apply sine activation
 4:      $a_2 \leftarrow \cos(z)$                                   ▷ Apply cosine activation
 5:      $output \leftarrow [a_1; a_2]$                              ▷ Concatenate activations
 6:      **return** $output$
 7: **end function**

---

## B   PROOFS

*Proof of Theorem 1.* We first present the result for two tasks and we then generalize it to an arbitary number of tasks. Let the linear weights learned on the first task be $\theta^{(T)}$, with the corresponding unique global minimum denoted by $\theta_1^\star$. The solution found on the first task is used as an initialization on the second task, which will end at $\theta^{(2T)}$, with the corresponding unique global minimum denoted by $\theta_2^\star$. We start from the known suboptimality gap for gradient descent on the second task (Garrigos and Gower, 2023):

$$J_2(\theta^{(2T)}) - J_2(\theta_2^\star) < \frac{\|\theta_2^\star - \theta^{(T)}\|^2}{\alpha T}. \tag{3}$$

We upper bound the distance from the initialization on the second task, $\theta^{(T)}$, to the optimum, $\theta_2^\star$, by

$$\|\theta_2^\star - \theta^{(T)}\|^2 < \|\theta_2^\star - \theta_1^\star\|^2 + \|\theta_1^\star - \theta^{(T)}\|^2 < \|\theta_2^\star - \theta_1^\star\|^2 + (1 - \alpha\mu)^T \|\theta_1^\star - \theta_0\|^2. \tag{4}$$

Where the last inequality uses the assumption that the objective function is $\mu$-strongly convex. We upper bound the suboptimality gap on the second task by a quantity independent of $\theta^{(T)}$:

$$J_2(\theta^{(2T)}) - J_2(\theta_2^\star) < \frac{\|\theta_2^\star - \theta^{(T)}\|^2}{\alpha T} < \frac{\|\theta_2^\star - \theta_1^\star\|^2 + (1 - \alpha\mu)^T \|\theta_1^\star - \theta_0\|^2}{\alpha T}, \tag{5}$$

which implies that the parameter value learned on the previous task does not influence training on the new task beyond a dependence on the initial distance. This is true for an arbitrary number of tasks:

$$J_\tau(\theta^{(\tau T)}) - J_\tau(\theta_\tau^\star) < \frac{\sum_{k=1}^\tau (1 - \alpha\mu)^{T(k-\tau)} \|\theta_k^\star - \theta_{k-1}^\star\|^2}{\alpha T} < \frac{2D(1 - \alpha\mu)^T}{\alpha T(1 - (1 - \alpha\mu)^T)}, \tag{6}$$

where we denote $\theta_0^\star = \theta_0$. The last inequality follows from our assumption that the distance between the task solutions, $\|\theta_k^\star - \theta_{k-1}^\star\|^2 < 2D$, is bounded and using a geometric sum in $(1 - \alpha\mu)^T$.   □

*Proof of Lemma 1.*   ⇒ We first prove the lemma in the forward direction:

Assuming that $\theta_{l,i}^{(t)} = \theta_{l',i}^{(t)}$ for $l' \neq l$, we will show that $\theta_{l,i}^{(t-1)} = \theta_{l',i}^{(t-1)}$.

Writing the gradient update for $\theta_{l,i}^{(t)}$ with a fixed step-size $\alpha$, we have that

$$\theta_{l,i}^{(t)} = \theta_{l,i}^{(t-1)} - \alpha \nabla_{\theta_{l,i}} J(\theta) \tag{7}$$

$$= \theta_{l,i}^{(t-1)} - \alpha \nabla_{f_\theta} \ell(f_\theta(x), y) \nabla_{\theta_{l,i}} f_\theta(x) \tag{8}$$

$$= \theta^{(t-1)} - \alpha \nabla_{f_\theta} \ell(f_\theta(x), y) \nabla_{\theta_{l,i}} \prod_{j=1}^L \theta_{j,i}^{(t-1)} x \tag{9}$$

$$= \theta_{l,i}^{(t-1)} - \alpha \nabla_{f_\theta} \ell(f_\theta(x), y) \prod_{j \neq l} \theta_{j,i}^{(t-1)} x. \tag{10}$$

$$\tag{11}$$

Similarly, the gradient update for $\theta_{l',i}$ is

$$\theta_{l',i}^{(t)} = \theta_{l',i}^{(t-1)} - \alpha \nabla_{f_\theta} \ell(f_\theta(x), y) \prod_{j \neq l'} \theta_{j,i}^{(t-1)} x \tag{12}$$

$$\tag{13}$$

Using our assumption that $\theta_{l,i}^{(t)} = \theta_{l',i}^{(t)}$, we set the two updates equal to eachother:

$$\theta_{l,i}^{(t-1)} - \alpha \nabla_{f_\theta} \ell(f_\theta(x), y) \prod_{j \neq l} \theta_{j,i}^{(t-1)} x = \theta_{l',i}^{(t-1)} - \alpha \nabla_{f_\theta} \ell(f_\theta(x), y) \prod_{j \neq l'} \theta_{j,i}^{(t-1)} x. \tag{14}$$

We can simplify both sides of the equations, where the LHS is

$$\theta_{l,i}^{(t-1)} - \alpha \nabla_{f_\theta} \ell(f_\theta(x), y) \prod_{j} \theta_{j,i}^{(t-1)} \frac{x}{\theta_{l',i}^{(t-1)}} \tag{15}$$

$$= \theta_{l,i}^{(t-1)} \left( 1 - \alpha \nabla_{f_\theta} \ell(f_\theta(x), y) \prod_{j} \theta_{j,i}^{(t-1)} \frac{x}{\theta_{l',i}^{(t-1)} \theta_{l,i}^{(t-1)}} \right). \tag{16}$$

Similarly, the RHS of the equation is

$$\theta_{l',i}^{(t-1)} \left( 1 - \alpha \nabla_{f_\theta} \ell(f_\theta(x), y) \prod_{j} \theta_{j,i}^{(t-1)} \frac{x}{\theta_{l',i}^{(t-1)} \theta_{l,i}^{(t-1)}} \right). \tag{17}$$

Notice that both expressions in the parenthesis on the LHS and RHS are equal. Thus, $\theta_{l',i}^{(t-1)} = \theta_{l,i}^{(t-1)}$
$\Leftarrow$ The reverse direction follows directly by following the above argument in reverse.

$\square$

*Proof of Lemma 2.* $\Rightarrow$ We first prove the lemma in the forward direction:
Assuming that $\bar{\theta}_i^{(t+1)} = \bar{\theta}_i^{(t)} = 0$, we will show that $\theta_{l,i}^{(t)} = \theta_{l',i}^{(t)} = 0$.

We proceed by contradiction, and assume that only a single component is zero, that is $\theta_{l',i}^{(t)} = 0$ and $\theta_{l,i}^{(t)} \neq 0$ for $l \neq l'$. We will show that the gradient update will ensure that $\theta_i^{(t+1)} \neq 0$

First, consider the update to $\theta_{l',i}^{(t)}$,

$$\theta_{l',i}^{(t+1)} = \theta_{l',i}^{(t)} - \alpha \nabla_{f_\theta} \ell(f_\theta(x), y) \prod_{j \neq l'} \theta_{j,i}^{(t-1)} x \tag{18}$$

$$= -\alpha \nabla_{f_\theta} \ell(f_\theta(x), y) \prod_{j \neq l'} \theta_{j,i}^{(t-1)} x \tag{19}$$

Because we assumed that $\theta_{l,i}^{(t)} \neq 0$ for $l \neq l'$, we have that $\prod_{j \neq l} \theta_{j,i}^{(t-1)} \neq 0$. Thus $\theta_{l',i}^{(t+1)} \neq 0$

Next consider the update to $\theta_{l,i}^{(t)}$,

$$\theta_{l,i}^{(t+1)} = \theta_{l',i}^{(t)} - \alpha \nabla_{f_\theta} \ell(f_\theta(x), y) \prod_{j \neq l} \theta_{j,i}^{(t-1)} x \tag{20}$$

$$= \theta_{l',i}^{(t)} \tag{21}$$

Where the last line follows from the fact that $\prod_{j\neq l} \theta_{j,i}^{(t-1)} = 0$ because $\theta_{l',i}^{(t)} = 0$.

Thus, we have shown that $\theta_{l,i}^{(t+1)} \neq 0$ for all $l$, and hence, $\bar{\theta}^{(t+1)} \neq 0$ which is a contradiction.

$\Leftarrow$ The reverse direction follows from the assumption directly. If two components are both equal to zero, $\theta_{l,i}^{(t)} = \theta_{l',i}^{(t)} = 0$, then every sub-product is zero, $\prod_{j\neq l} \theta_{j,i}^{(t-1)}$ and so is the entire product, $\prod_{j=1}^{L} \theta_{j,i}^{(t-1)}$. $\qquad\square$

*Proof of Theorem 2.* We now show that a diagonal linear network maintains a positive minimum singular value under continual learning. This is a simplified setting for analysis, where we assume that the weight matrices are diagonal and thus the input, hidden, and output dimension are all equal. Let $f_\theta(x)$ be a diagonal linear network, defined by a set of diagonal weight matrices, $\theta_l = \text{Diag}(\theta_{l,1}, \dots, \theta_{l,d})$. The output of the diagonal linear network is the product of the diagonal matrices, $f_\theta(x) = \theta_L \theta_{L-1} \dots \theta_1 x$. Then the product matrix is also a diagonal matrix, whose diagonals are the products of the parameters of each layer, $\bar{\theta} = Diag(\prod_{l=1}^{L} \theta_{l,1}, \dots, \prod_{l=1}^{L} \theta_{l,d}) := Diag(\bar{\theta}_1, \dots, \bar{\theta}_d)$. The minimum singular value of a diagonal matrix is the minimum of its absolute values, $\sigma_{min}(\bar{\theta}) = \min_i |\bar{\theta}_i|$. Thus, we must show that the minimum absolute value of the product matrix is never zero.

This follows immediately from Lemma 1 and Lemma 2. Taken together, these two lemmas state that with a random initialization and under gradient dynamics, a diagonal linear network will not have more than one parameter equal to zero. This means that the minimum singular value of the product matrix will never be zero. Thus, we have shown that a diagonal linear network trained with gradient descent, if initialized appropriately, will be able to converge on any given task in a sequence. $\qquad\square$

*Proof of Proposition 1.* We prove this by considering the remainder of a Taylor series on the given interval. Due to periodicity of $\sin(z)$ and $\cos(z)$, we can consider $z \in [-\pi, \pi]$ without loss of generality. We can further consider two cases, either $z \in [-\pi, {-3\pi}/{4}] \cup [{-\pi}/{4}, {\pi}/{4}] \cup [{3\pi}/{4}, \pi]$ or $h \in [{-3\pi}/{4}, {-\pi}/{4}] \cup [{\pi}/{4}, {3\pi}/{4}]$. In the first case, $z$ is near a critical point of $\cos(z)$ and in the second case $z$ is near a critical point of $\sin(z)$.

We focus on a particular subcase, where $z \in [{-\pi}/{4}, {\pi}/{4}]$, which is close to a critical point of $\cos(z)$, but far from a critical point of $\sin(h)$ (the other cases follow a similar argument).

Because we know that $z \in [{-\pi}/{4}, {\pi}/{4}]$, by Taylor's theorem it follows that $\sin(z) = z + R_{1,0}(z)$, where $R_{1,0}(z) = \frac{\sin^{(2)}(c)}{2} z^2$ is the 1st degree Taylor remainder centered at $a = 0$ for some $c \in [{-\pi}/{4}, {\pi}/{4}]$. In the case of a sinusoid, this can be upperbounded, $|R_{1,0}(z)| = |\frac{-\sin(c)}{2} z^2| < \frac{1}{8\sqrt{2}} (\pi/4)^2$, using the fact that $|z| < \pi/4$ and $\sin(c) < 1/\sqrt{2}$.

Thus, when $\cos(z)$ is close to a critical point, $\sin(z)$ is approximately linear. A similar argument holds for the other case, when $\sin(z)$ is close to a critical point, $\cos(z)$ is approximately linear. In this other case, the error incurred is the same.

$\qquad\square$

*Proof of Corollary 1.* We prove this claim using induction.

Base case: We want to show that a single layer that outputs Fourier features embeds a deep linear network. Using Proposition 1, there exists one unit for each pre-activation that is approximately linear. Because each pre-activation is used in an approximately-linear unit, the single layer approximately embeds a deep linear network using all of its parameters.

Induction step: Assume a deep Fourier network with depth $L - 1$ embeds a deep linear network, we prove that adding an additional deep Fourier layer retains the embedded deep linear network. There are two cases to consider, corresponding to the units of the additional deep Fourier layer which are approximately-linear and the other units that are not approximately-linear

Case 1 (approximately-linear units): For the additional deep Fourier layer, the set of approximately-linear units already embeds a deep linear network. Because linearity is closed under composition, the composition of the additional deep Fourier layer and the deep Fourier network with depth $L - 1$ simply adds an additional linear layer to the embedded deep linear network, increasing its depth to $L$.

Case 2 (other units): For the units that are not well-approximated by a linear function, we can treat them as if they were separate inputs to the deep Fourier network with depth $L - 1$. The network's parameters associated with those inputs are, by the inductive hypothesis, already embedded in the deep linear network.

Note that case 1 embeds the parameters of the additional deep Fourier layer into the deep Fourier network. Case 2 states that the parameters of the network associated with the nonlinear units of the additional deep Fourier layer are already embedded in the deep Fourier network by construction.

Thus, a neural network composed of deep Fourier layers embeds a deep linear network. □

## C EMPIRICAL DETAILS

All of our experiments use 10 seeds and we report the standard error of the mean in the figures. The optimiser used for all experiments was Adam, and after a sweep on each of the datasets over $[0.005, 0.001, 0.0005]$, we found that $\alpha = 0.0005$ was most performant.

We used the Adam optimizer (Kingma and Ba, 2015) for all experiments, settling on the default learning rate of $0.001$ after evaluating $[0.005, 0.001, 0.0005]$. Results are presented with standard error of the mean, indicated by shaded regions, based on 10 random seeds.

Dataset specifications and non-stationarity conditions:

- For MNIST, Fashion MNIST and EMNIST: we use a random sample of 25600 of the observations and a batch size of 256 (unless otherwise indicated, such as the linearly separable experiment).

- For CIFAR10 and CIFAR1100: Full 50000 images for training, 1000 test images for validation, rest for testing. The batch size used was 250. Labelnoise non-stationarity: 60 epochs, 10 tasks. Class incremental learning: 6000 iterations per task, 80 tasks. Note that the datasets on different tasks in the class incremental setting can have different sizes, and so epochs are not comparable.

- tiny-ImageNet: All 100000 images for training, 10000 for validation, 10000 for testing as per predetermined split. The batch size used was 250. Labelnoise experiment non-stationarity: 80 epochs per task, 10 tasks total. Class incremental learning: 10000 iterations per task, 80 tasks. Note that the datasets on different tasks in the class incremental setting can have different sizes, and so epochs are not comparable.

**Neural Network Architectures** For tiny-ImageNet, CIFAR10, CIFAR100, and SVHN2: We utilized standard ResNet-18 with batch normalization and a standard tiny Vision Transformer. The smaller datasets use an MLP with different widths and depths, as specified in the scaling section.

**Experiment Metrics** All figures reporting accuracy evaluate the accuracy on the distribution given by the current task. Figures 2, 3, 4 and 7 (top) report the training accuracy on the current task. Figures 5 and 6 report the test accuracy on the current task. Figure 5 shows the accuracy at the end of each epoch, whereas Figure 6 shows the accuracy at the end of each task (due to too many tasks). The accuracy reported in Figure 7 is the final accuracy at the end of the last task.

# D ADDITIONAL EXPERIMENTS

## D.1 ADDITIONAL DEEP LINEAR NETWORK RESULTS

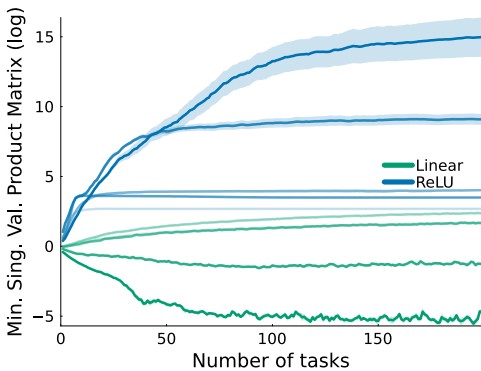

Figure 8: **Minimum singular value of the product matrix for a deep general linear network on a linearly-separable task.**

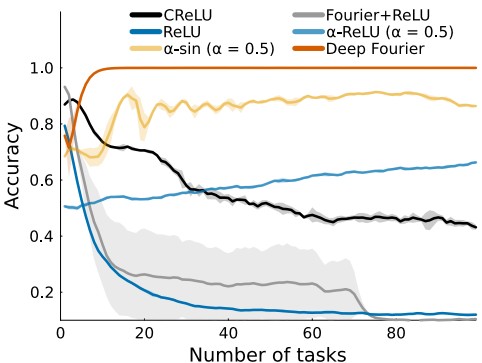

Figure 9: **Average unit sign entropy on a non linearly-separable task.** Deep Fourier features and other perioidic activation functions have high average unit sign entropy compared to piecewise linear activations like `ReLU` and `leaky-ReLU`.

## D.2 ABLATING ARCHITECTURE WITH WIDE RESIDUAL NETWORKS

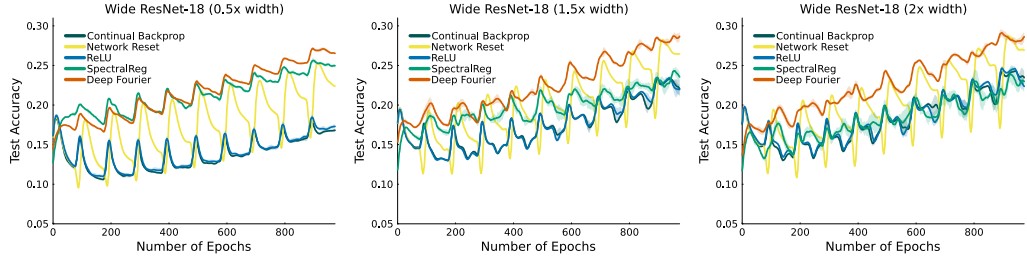

Figure 10: **Investigating Wide Residual Networks with different width scales on tiny-ImageNet with label noise.**

## D.3 INVESTIGATING SINGLE TASK PERFORMANCE

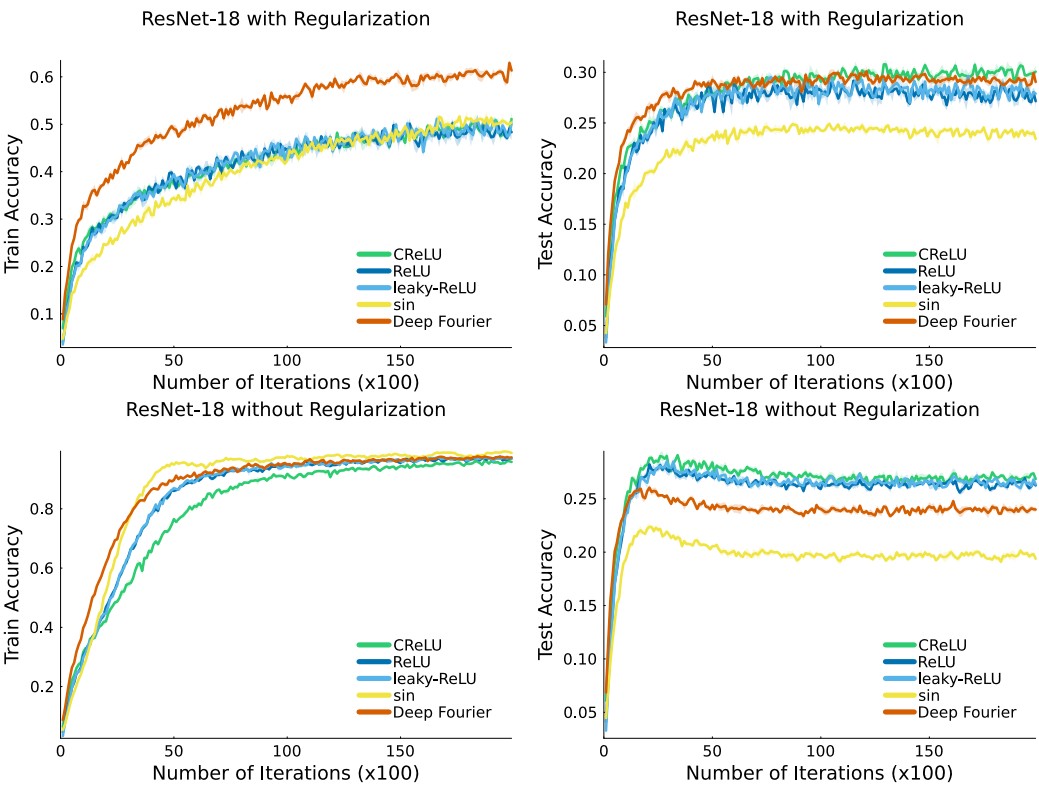

Figure 11: **Investigating single task performance of ResNet-18 using different activation functions on tiny-ImageNet.**

## D.4 CONTINUAL IMAGENET RESULTS

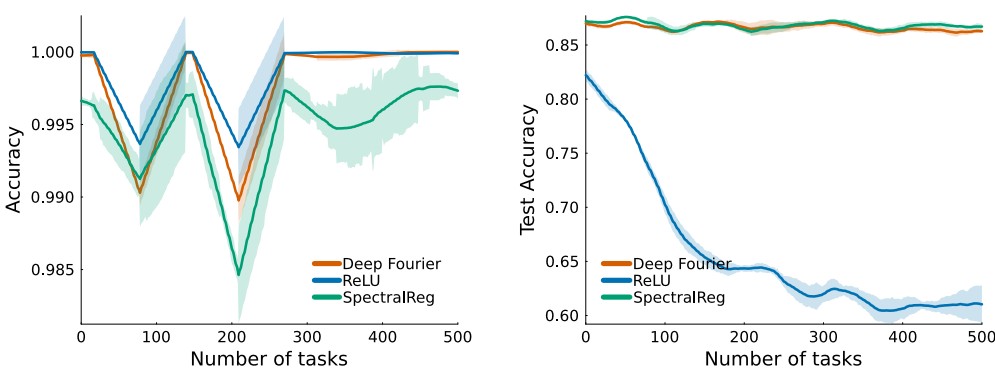

Figure 12: **Investigating sustained performance on many tasks using ResNet-18 on Continual ImageNet.** Note that loss of trainability does not occur on this problem, whereas loss of generalization does occur.

## D.5 COMPARING CONTINUAL BACKPROP TO RECYCLING DORMANT NEURONS

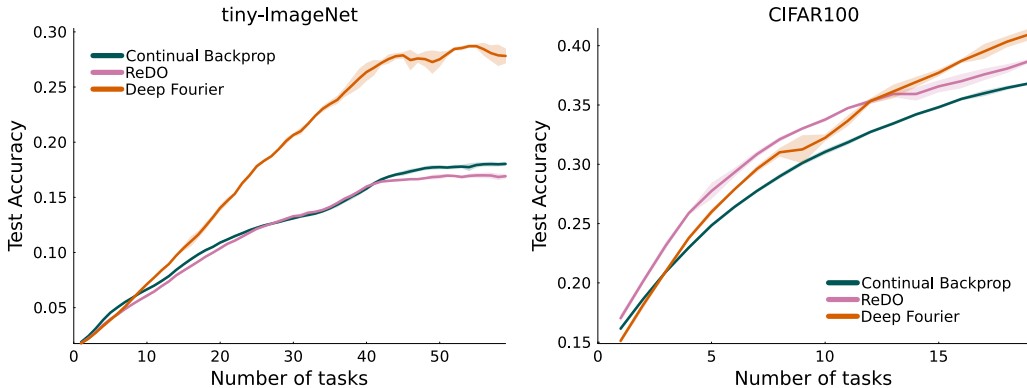

Figure 13: **Recycling dormant neurons and continual backprop are both weight reinitialization methods that perform similarly.**

## D.6 ADDITIONAL SENSITIVITY RESULTS

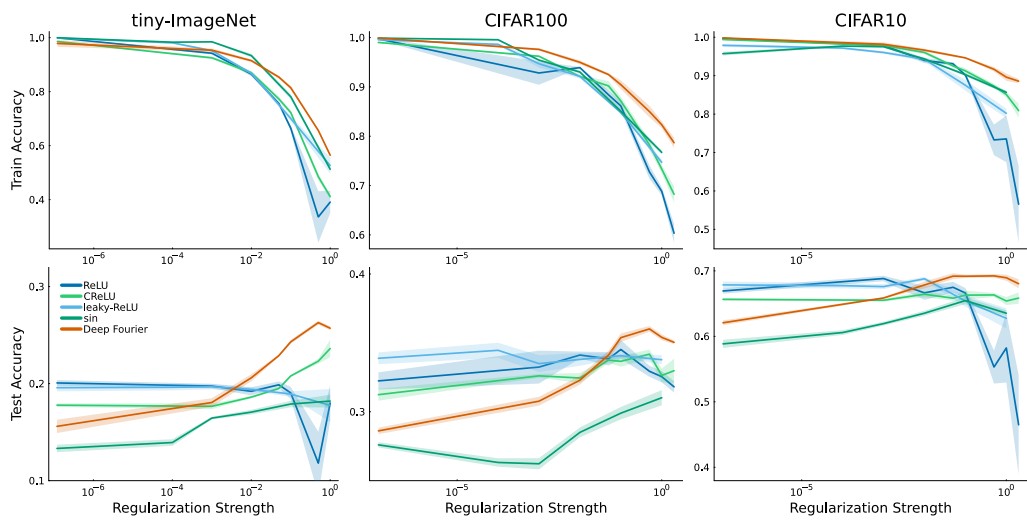

Figure 14: **Sensitivity analysis on tiny-ImageNet, CIFAR10, and CIFAR100.** Networks with deep Fourier features are highly trainable, but have a tendency to overfit without regularization, leading to high training accuracy but low test accuracy. Due to deep Fourier features being highly trainable, they are able to train with much higher regularization strengths leading to ultimately better generalization.

## D.7 Forgetting Results

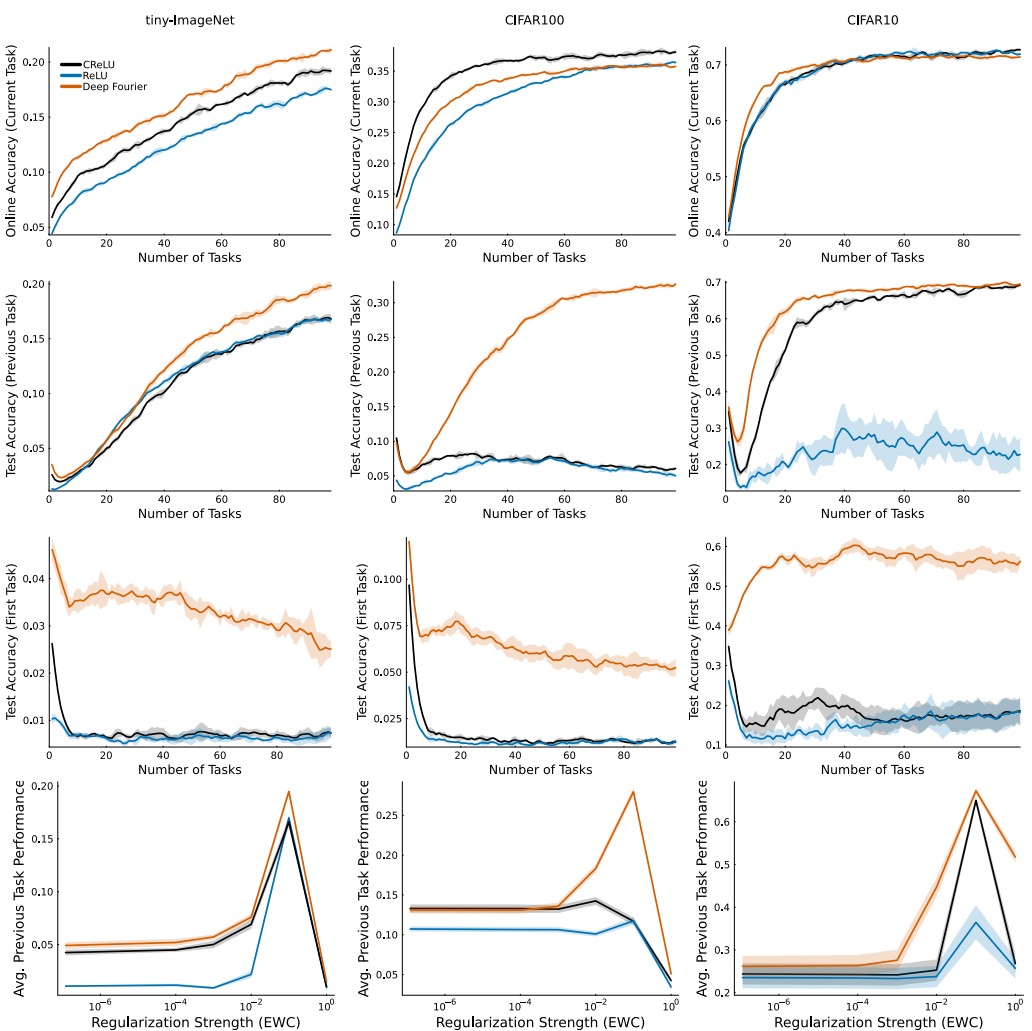

Figure 15: **Forgetting on online label-permuted tiny-ImageNet, CIFAR10, and CIFAR100.** All networks are capable of continual online learning within a task, indicating that they maintain plasticity and succeed in avoiding catastrophic forgetting data early within a single task. However, deep Fourier features are particularly capable of maintaining performance on previous tasks.

## D.8 Additional Trainability Results Using Deep Fourier Features

These additional experiments validate the benefits of deep Fourier features as a means of improving trainability. The experiments use the following datasets for continual supervised learning: MNIST (LeCun et al., 1998), Fashion MNIST (Xiao et al., 2017), and EMNIST (Cohen et al., 2017). We focus primarily on the problem of trainability, and thus consider random label non-stationarity, in which the labels are randomly assigned to each observation and must be memorized on each task. This type of non-stationarity is particular difficulty in sustaining trainability in continual learning (Lyle et al., 2023; Kumar et al., 2023b). We compare our network with deep Fourier feature against a corresponding feed-forward neural network with `ReLU` activations with the same depth. Because deep Fourier features use a concatenation of two different activation functions, it has half the width of the `ReLU` network and less parameters, which provides an advantage to the `ReLU` baseline.

### D.8.1 DEEP FOURIER FEATURES ARE HIGHLY TRAINABLE

The main result of this appendix is presented in Figure 16. Across different datasets, deep Fourier feature networks are highly trainable, either achieving high accuracy and maintaining it on easier tasks, such as MNIST, or improving their trainability on new tasks, such as on Fashion MNIST. In contrast, the `ReLU` network suffered from loss of trainability in each of the problems that we studied. This is not surprising, as loss of trainability is a well-documented issue for `ReLU` networks without some additional method designed to mitigate it (Dohare et al., 2021; Lyle et al., 2022; Kumar et al., 2023b; Elsayed and Mahmood, 2024).

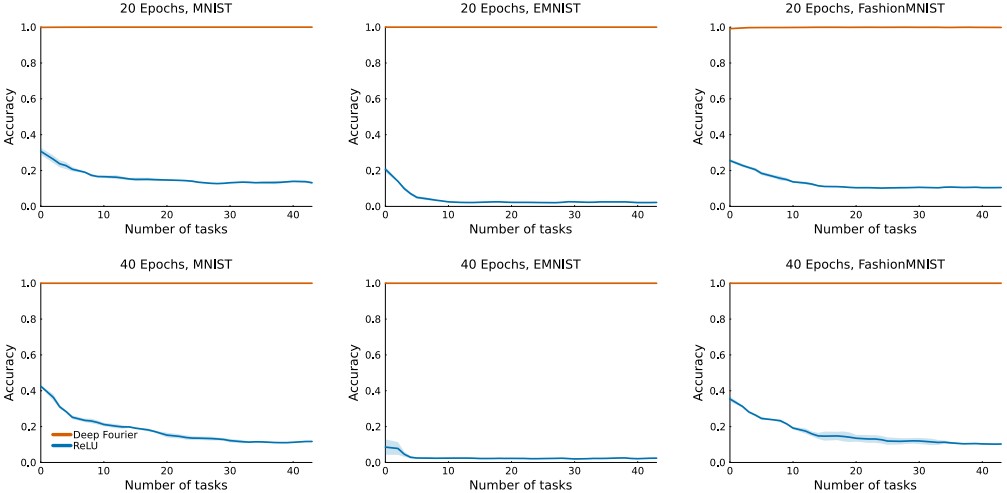

Figure 16: **Trainability across different datasets and epochs per tasks.** `ReLU` networks lose their trainability, whereas networks with deep Fourier features improve and sustain their trainability

### D.8.2 METHODS FOR IMPROVING TRAINABILITY

Given that a `ReLU` network is unable to maintain its trainability in isolation, we investigate whether recently proposed methods for mitigating loss of trainability are able to make up for the difference in performance between a network with deep Fourier features and a network with `ReLU` activations. We investigate two categories of mitigators for loss of plasticity: (i) regularization and (ii) normalization layers.

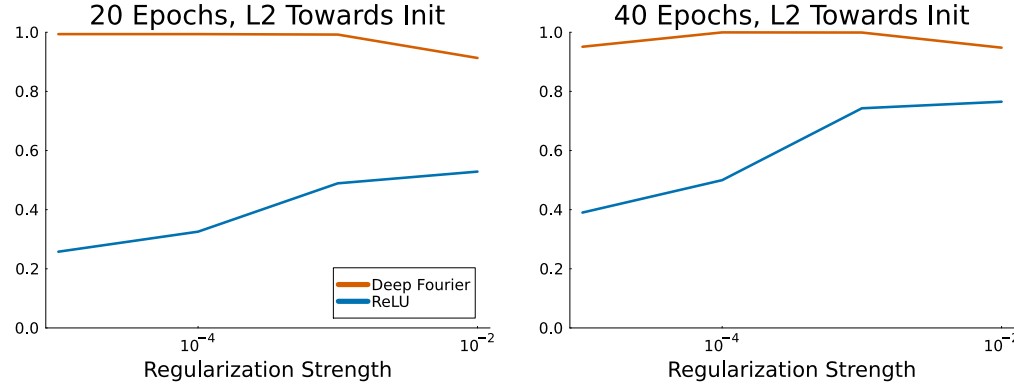

Figure 17: **Hyperparameter Sensitivity Analysis.** Deep Fourier features seem to not benefit from regularization for trainability. While `ReLU` networks are more trainable with regularization, their performance is still worse than the deep Fourier feature network. Note that the experiments in Section 5.2 indicate that deep Fourier features do benefit from regularization for generalization.

**Regularization** Loss of plasticity occurs in `ReLU` networks when they are not regularized. Thus, we compare the performance of the `ReLU` network and the deep Fourier feature network with varying regularization strengths. In particular, we use the recently proposed L2 regularization towards the initialization (Kumar et al., 2023b), because it addresses the issue of sensitivity towards zero common to L2 regularization towards zero. In Figure 17, we find that regularization does improve the trainability of `ReLU` networks, validating previous empirical findings. However, we found that deep Fourier feature networks do not benefit substantially from regularization. That is, deep Fourier feature network with a smaller regularization strength always outperformed the `ReLU` network.

**Layer Normalization** Training deep neural networks typically involve normalization layers, either Batch Normalization (Ioffe and Szegedy, 2015) or Layer Normalization (Ba et al., 2016). Recently, it was demonstrated that layer normalization is an effective mitigator for loss of trainability (Lyle et al., 2024). We investigate whether trainability can be improved with the addition of normalization layers, for both the `ReLU` and deep Fourier feature network. In Figure 18, we found that layer normalization increases performance but that loss of trainability can still occur with a `ReLU` network. In addition to Layer Normalization, we also tried a linear version of LayerNorm which uses a stop-gradient on the standard deviation to maintain linearity, which improved training speed in some instances.

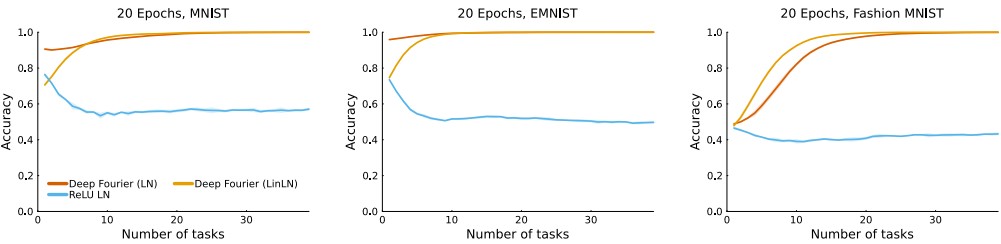

Figure 18: **Comparison of trainability with Layer Normalization.** `ReLU` networks are more trainable with Layer Normalization, but deep Fourier feature networks learn faster and achieve better accuracy, particularly with linearized Layer Norm.

### D.8.3 SCALING PROPERTIES OF DEEP FOURIER FEATURE NETWORKS

Figure 19: **Scaling Neural Network Width and Depth.** (Top) Due to the concatenation used by the activation function in deep Fourier feature networks, they scale particularly well with width. (Bottom) Deeper Fourire features also lead to improved average end of task performance.

**Width Scaling**  Another source of linearity recently proposed is an increasing width of the neural network, causing their parameter dynamics evolves as linear models in the limit (Lee et al., 2019). We investigate whether an increase in width can close the gap between the trainability of the `ReLU` network and the deep Fourier feature network. In Figure 19 (Top), we found that deep Fourier feature networks scale particularly well with width, whereas width seems to have little effect on the trainability of `ReLU` networks. Thus, our results suggest that increasing the width of a neural network does not necessarily impact its trainability, at least not to the width values we considered.

**Depth Scaling**  Neural networks in supervised learning tend to scale with depth, allowing them to learn more complex predictions. We investigate whether the depth scaling of deep Fourier feature networks also leads to similar improvements in continual learning. In Figure 19 (Bottom), we found that deep Fourier feature networks do improve with additional depth, but the degree of improvement was not as pronounced as scaling the width.

