# OpenReview forum: "Plastic Learning with Deep Fourier Features"
_ICLR.cc/2025/Conference — ICLR 2025 Poster_

### Official Review · Reviewer_tK5u · 2024-10-28

**Soundness:** 3
**Presentation:** 4
**Contribution:** 3
**Rating:** 6
**Confidence:** 3

**Summary:**

This paper studies the loss of plasticity in commonly faced by deep neural networks in continual learning settings: As the training data continues to shift, eventually most deep networks start to lose their ability to adapt to the new distributions. The authors observe empirically and provide theoretical analysis showing that deep linear networks do not suffer this same problem. However, deep linear networks do not have the same representation power as nonlinear ones, so the authors instead propose imbuing nonlinear networks with more linear characteristics through deep Fourier features. The paper goes on to explain why alternative activation functions fail to embed a similarly linear network, and empirically demonstrate how deep Fourier features are able to preserve model trainability in two continual learning settings (Diminishing Label Noise, a variant of Class-Incremental Learning), for CIFAR-10, CIFAR-100, and tiny-ImageNet.

**Strengths:**

S1. A well-motivated, principled perspective: I’ve been reading (and reviewing) papers on continual learning for 7 years now, and most of the papers I’ve seen after the first papers that established the popular continual learning paradigms (e.g. replay, regularization, parameter isolation/expansion) have continued to fall into those buckets. Generally, many of the submissions I see are hacky and heuristics driven, consisting of various remixes of previously explored ideas. I thus found this paper to be a breath of fresh air, examining some of the problems of continual learning from a more theoretical perspective, building upon past works on plasticity, activations functions, Fourier features, and more to examine what exactly about nonlinear deep networks leads to lower plasticity for future tasks, as well as how to mitigate. I want to see more papers trying to solve continual learning from this kind of approach.

S2. Insights on plasticity: Continual learning papers tend to focus on the catastrophic forgetting problem, forgetting that being able to continually learn is also part of the problem. This paper doesn’t originate the observation of the loss of trainability in nonlinear deep networks (previous works cited), so it shouldn’t be attributed to the authors, but I would like to see more activity from the continual learning on this topic. I feel this paper could make a good contribution in this sense.

S3. Connections with other common activations: I appreciate the framing of other common activation types like ReLU and leaky ReLU from the perspective of this paper’s analysis (especially lines 361-369). These were helpful for understanding why these standard practices might lose trainability, and why the proposed Fourier feature approach might succeed.

S4. Writing: The paper is written very clearly and are organized well. The small empirical experiments in Figures 2 and 4 very compactly provides good empirical evidence of trainability over long task sequences for linear and non-linear models. Thanks for including.

**Weaknesses:**

W1. What about catastrophic forgetting?/Why not re-init?: A discussion about catastrophic forgetting--the most prominent challenge of continual learning--is largely absent from the paper. To be fair, trainability/plasticity is also often absent from papers focusing on catastrophic forgetting, and it’s good to have a deeper exploration on trainability given the lesser attention to it. However, I still feel a discussion and empirical investigation of deep Fourier features response catastrophic forgetting behavior is necessary; focusing on just trainability doesn’t make much sense to me, because if trainability is a major problem, then worst case a model can just be re-initialized and retrained from scratch. See Q4.  If the problem can be solved by just re-initialization, then the contribution here is interesting but not practical. This is currently my biggest concern.

W2. Strong Assumptions: While I understand that strong assumptions are commonplace in many theoretical treatments of deep learning, the restrictiveness of some of the assumptions here still are worth calling out. In particular, Theorem 2 assumes a deep diagonal linear network. Despite the cited precedents for such an assumption, this is pretty far from the reality of my deep networks in practice. How much do these insights shed here transfer to the more general deep linear network setting?


W3. Double width vs half representation: The proposed method involves concatenating sin(z) and cos(z). This either involves double the width of the network (increases parameters and computation time) or halves the effective width (reduced model capacity). For a fixed size head-to-head comparison then, models parameterized this way may be at a sizable disadvantage. The experiments however are fair in this regard however, as they choose to incur the half effective penalty rather than unfairly double model width.

W4. Page Length: The page limit of ICLR now permits papers up to 10 pages, but 9 pages is highly encouraged; the recommendation for using the tenth page is more for including larger and more detailed figures. This paper extends past 9.5 pages, but I personally do not feel the visual content justifies going over the 9-page recommendation. I would recommend the authors try to get down to 9 pages (e.g. moving more content to the Appendix), or else for fairness reasons I believe this paper should be evaluated at a higher bar.

Miscellaneous:
- Figure 3: Having a graph inset in another like is non-standard and can be a little confusing. It seems like this may be better suited as two separate subplots.
- Linear 273: “function” => “functions”
- Line 419: Class incremental learning definition: This is not the traditional class incremental learning setup usually shown in most continual learning papers. Notably, it seems like addressing catastrophic forgetting is completely optional, as new tasks still contain the classes from the previous tasks. I would suggest naming this something else to distinguish it from the more common (and harder) setting.
- Line 465: extra space before footnote
- Figure 6 (right) caption: It doesn’t appear that deep Fourier features “substantially improve over most baselines”. In fact, they seem pretty middle of the pack?

**Questions:**

Q1: Line 117: “The problem setting that we consider is continual supervised learning without task boundaries.” The description that follows sounds like there are task boundaries every $T$ iterations? Is the “without” a typo?

Q2: Figure 4: Any reason not to also include plain ReLU in this figure? I presume it does badly.

Q3: Figure 5: Deep Fourier features have a very stark advantage over the other approaches for tiny-ImageNet, but the sizable gap disappears on less complex datasets like CIFAR 10 and 100. This happens in Figure 6 too. Why do you suspect this is?

Q4: In both the experimental settings, it doesn’t seem like any sort of remembering of past tasks is necessary. For diminishing noise, the label space remains the same, and the new data distributions just contain better information; for the class-incremental setting, the new data distributions contain all past distributions as a subset. What about just re-initializing the model and training from scratch? It would be good to have this naïve baseline also plotted in Figure 5 + 6.

---

> ### Author Response · Authors · 2024-11-24
>
> Thank you for your insightful comments, we particularly appreciate you referring to the paper “as a breath of fresh air.” The updated submission includes several new results and clarifying sentences (all changes and new content is in blue, except for minor typos which have been corrected). We address your comments below,
> Weaknesses:
>
> > A discussion about catastrophic forgetting--the most prominent challenge of continual learning--is largely absent from the paper. To be fair, trainability/plasticity is also often absent from papers focusing on catastrophic forgetting, and it’s good to have a deeper exploration on trainability given the lesser attention to it. [...] worst case a model can just be re-initialized and retrained from scratch.
>
> Forgetting and plasticity: As you mention, this paper is specifically about the more recently established loss of plasticity problem. We have added some additional discussion around forgetting in Section 1 (see “on forgetting:” in the shared reply).
>
> On reinit: You raise a good point. Note that we are considering a much harder problem setting where the learner has no information about a task change (see “on task-boundary:”). Thus, a reinitialization method is not applicable because knowing when to reinitialize requires knowing when the task changes. Notice that this is in fact one of the most appealing aspects of our proposed method: our results using deep Fourier features show that expensive reinitialization is not necessary to sustain good performance, this additional information is not even required!
>
> A similar “why not reinit” argument can be applied to the catastrophic forgetting setting, and continual learning more generally: a strong but computationally infeasible baseline could store all past data and retrain a new network on every new datapoint (e.g., see [1]). This has motivated recent works on memory and compute limitations as part of the continual learning problem formulation.
>
> [1] Prabhu et al. (2020). Gdumb: A simple approach that questions our progress in continual learning. ECCV.
>
> >W2. [...] Theorem 2 assumes a deep diagonal linear network. [...] How much do these insights shed here transfer to the more general deep linear network setting?
>
> You raise a good point, and although our theory was developed in the same settings as other recent papers that we referenced, we share the reviewer’s concerns. This is exactly why we provide, in Figure 2, experimental evidence showing that our trainability result holds for general deep linear networks. The theoretical analysis we can perform is limited by the existing techniques in the field (and it was beyond the scope of our paper to develop a theoretical framework for this), and that’s exactly the role of the empirical results in our paper: to empirically validate our insights when the assumptions we make for our theoretical results do not hold. If the theory was able to perfectly capture our empirical methodology, we wouldn’t necessarily need to run experiments, as the theoretical results would support themselves.
>
> >W3. Double width vs half representation: [...] The experiments however are fair in this regard however, as they choose to incur the half effective penalty rather than unfairly double model width.
>
> We are glad that the reviewer acknowledges that we are actually giving the baselines an advantage relative to the half-width of deep Fourier features. However, we wonder why the reviewer considers this to be a weakness. Please also see the new results in Appendix D.2. where we consider Wide Residual Networks to investigate the effect of width.
>
> >W4. Page Length:
>
> Thanks for raising this point, we have moved some of the extended discussion on deep linear networks to the appendix so that the paper fits in 9 pages.
>
>
> >Line 419: Class incremental learning definition: This is not the traditional class incremental learning setup usually shown in most continual learning papers. Notably, it seems like addressing catastrophic forgetting is completely optional, as new tasks still contain the classes from the previous tasks. I would suggest naming this something else to distinguish it from the more common (and harder) setting.
>
> We use the nomenclature introduced by [1], in which they say class-incremental learning involves a learning algorithm that must “learn to discriminate between a growing number of classes”. Also note that our class incremental experiment is equivalent to the class-incremental experiment in [2]. In both their experiments and in ours, each task corresponds to a distribution over all classes seen so far, which is well-represented by “a growing number of classes.” We have added a clarifying sentence to Section 5.1 to reflect this.
>
> [1] Van de Ven et al. (2022). Three types of incremental learning. Nature Machine Intelligence (4).
>
> [2] Dohare et al. (2024). Loss of plasticity in deep continual learning. Nature (632).
>
> [3] Lyle et al. (2023). Understanding Plasticity in Neural Networks. ICML.

---

> > ### Author Response · Authors · 2024-11-24
> >
> > >Figure 6 (right) caption: It doesn’t appear that deep Fourier features “substantially improve over most baselines”.
> >
> > In Figure 6 (right), the network with deep Fourier features is the second best performer. Our choice of color for the CReLU baseline may have caused this confusion. We have updated the color to make this result more clear.
> >
> > >Q1: Line 117: “The problem setting that we consider is continual supervised learning without task boundaries.” The description that follows sounds like there are task boundaries every iterations? Is the “without” a typo?
> >
> > On task-boundary: Thanks for bringing up this point: the task boundary information is not provided to the learner. This is the typical setting in the loss of plasticity literature [2,3]. We can clarify this further: the distribution changes every T iterations but the learner only receives samples from this now-different distribution, without any separate variable indicating the distribution has changed.  We have edited this sentence to be more clear.
> >
> > >Q2: Figure 4: Any reason not to also include plain ReLU in this figure? I presume it does badly.
> >
> > The reviewer is correct, the plain ReLU network does poorly as demonstrated in the earlier figures. We wanted to minimize clutter in this already busy plot, and to highlight the fact that shallow Fourier features, in combination with ReLU activations, are unable to mitigate loss of trainability. We have edited the figure to reduce clutter and make room for the plain ReLU baseline.
> >
> > >Q3: Figure 5: Deep Fourier features have a very stark advantage over the other approaches for tiny-ImageNet, but the sizable gap disappears on less complex datasets like CIFAR 10 and 100. This happens in Figure 6 too. Why do you suspect this is?
> >
> > Thanks, that’s an excellent question on which we can provide speculation. Our basic intuition is that more complex tasks take more iterations to learn, and that loss of plasticity is more likely to occur during the longer learning process. Compared to CIFAR10/CIFAR100, more iterations are required to learn a classifier on the tiny-imagenet dataset because it has larger resolution images (64x64 vs 32x32), more classes (200 vs 10/100) and more datapoints (100k vs 50k). We saw in Figure 1 that, for a given continual learning problem, decreased depth (as well as width in other work [2]) can magnify loss of trainability. We speculate that for a given architecture, more complex tasks that require more iterations to learn have a similar effect in magnifying loss of plasticity. Thus, the performance difference between methods becomes magnified on these more complex continual learning problems. We added a sentence to clarify this at the end of Section 5.1
> >
> > >Q4:In both the experimental settings, it doesn’t seem like any sort of remembering of past tasks is necessary. [...] What about just re-initializing the model and training from scratch? It would be good to have this naïve baseline also plotted in Figure 5 + 6.
> >
> > The reviewer is correct that remembering past tasks is not part of the current experiments. See also “On reinit:”
> >
> >  That being said, we have added this additional reset as a type of pseudo-oracle baseline to better compare the results achieved by deep Fourier features, which you can see in Figure 10 in Appendix D.2. We will also include it in the reset of our experiments.

---

> ### Comment · Reviewer_tK5u · 2024-11-24
> **Responses to Author rebuttal**
>
> I thank the authors for the effort they put into their responses, including updating the paper  (including getting under 9 pages) and adding experiments. I read through the other reviews when they were first released, as well as the authors’ responses.
>
> **Plastic Learning vs Re-Init**: The authors’ response is helpful, and I’m a little more open to the value of studying plastic learning on its own now, but I still have some remaining doubts:
> * The proposed method not requiring knowledge of task boundaries is a good point and a valid advantage. On the other hand, task-free continual learning methods [a,b] also deal with this problem, so there are existing solutions. Additionally, I wonder if even more basic solutions like outlier detection or simply monitoring for decreased training efficiency or lower accuracy would also solve the problem.
> * I disagree that the “why not reinit” argument can be similarly applied to catastrophic forgetting. Most continual learning settings explicitly do not allow storing all the data from past tasks (the replay class of methods store a subset, but I personally view this as cheating—but I digress). Thus, the brute force solution you propose is generally not an allowable solution. In contrast, I see no reason why re-init is not a valid solution for plasticity; with pre-trained backbones (especially with recent advances in self-supervised learning or growing availability of “foundation” models), it may not even be that expensive to do.
>
> This remains one of my largest concerns. Without a resolution, I’m hesitant on whether plasticity, without being coupled with catastrophic forgetting, is by itself a topic worth solving.
>
> **Strong assumptions**: Fair, point taken. On the other hand, I would modify the text a bit to make it clear that these are strong assumptions, and that while the theory doesn’t prove that modern neural networks also behave similarly, you provide empirical results indicating that deep Fourier features might. For example, the paragraph right after Section 3 makes the connection between the theory and the more general case seem a little to easy.
>
> **Why is half-representation a weakness?**: The experiments are fair, but that doesn’t change the fact that given a model size budget, this method halves the effective width of the representation. For the experiments in this paper, this didn’t turn out to be a problem, but in other settings, it could be. For example:
> * I suspect this choice may be a problem for more complex data distributions or learning problems than CIFAR or tiny Imagenet, where the model is less over-parameterized and actually needs all the representational power it can get.
> * Going in the opposite direction, my guess is that halving of the representation for the same number of parameters negatively impacts one’s ability to compress the model, which is often important when deploying a model in the real world.
>
> I thus stand by my characterization that this is a weakness, but I will note that it didn’t factor that significantly in my score.
>
>
> [a] Aljundi et al. “Task-Free Continual Learning”. CVPR 2019 \
> [b] Lee et al. “A Neural Dirichlet Process Mixture Model for Task-Free Continual Learning” ICLR 2020

---

> > ### Author Response · Authors · 2024-11-27
> >
> > We truly appreciate the reviewer’s engagement in the rebuttal period.
> >
> >
> > We now have updated the submission with several new results demonstrating deep Fourier features can also provide benefits in avoiding catastrophic forgetting (see Appendix D.7). Our experiment uses the setting described by [4], in which the labels for a dataset are permuted at the beginning of each task, and learning is online, meaning that each data point is seen exactly once. Moreover, and unlike the experiments in [4], we use more large-scale datasets (+CIFAR100/tiny-ImageNet) and train a ResNet end-to-end rather than just the last two layers. A good continual learning algorithm on this problem should both i) improve as more tasks are seen and ii) maintain performance on previously seen tasks. For all methods considered (ReLU, CReLU and deep Fourier features), we use elastic weight consolidation to regularize towards the parameters of the previous task [5]. Our results demonstrate that deep Fourier features can sustain high accuracy on the current and previous task, and that deep Fourier features are robust to the hyperparameter strength across online label-permuted tiny-ImageNet, CIFAR10 and CIFAR100.
> >
> >
> > We thank the reviewer for pushing us to do so, as this makes our submission more complete. The reason we did not do it in the first place is that we do think that loss of plasticity is "by itself a topic worth solving.”
> >
> >
> > Thus, we would also like to convince the reviewer of the importance of plasticity as a standalone topic. The reviewer points out that storing past data does not address the catastrophic forgetting problem (the reviewer refers to this as “cheating”). The same is true for loss of plasticity; a full reset of the network does not address this problem. A full network reset is both i) ineffective in leveraging previously learned representations and ii) inefficient because of the computation cost of training after resetting, especially for large models.
> >
> > However, storing past data or resetting a network could be valid in certain situations. Perhaps past data is pertinent and available, or a network has reached an unrecoverable state. Thus, we should acknowledge the underlying trade-offs for storing data (which costs memory) or resetting a network (with costs computation to retrain).
> >
> > Notice that:
> >
> > - Other authors have recently argued that the cost of computation is much higher than the cost of memory [1]. Accordingly, many have started to consider the computationally-constrained setting for continual learning [2,3]. This would imply that an efficient approach to continual learning would tend to store more data and reset the network less often.
> >
> > - In our updated results, deep Fourier features already surpass the full reset baseline. This is because our proposed network with deep Fourier features learns a representation from the previous task, which is helpful for the next task. Our method also beats selective reset baselines like ReDO and continual backprop, which both use heuristics for resetting specific parameters. Thus, our method is both effective and efficient as demonstrated by the results in the problems that we consider.
> >
> >
> > We hope this sufficiently convinces the reviewer and that they will consider updating their score.
> >
> > [1] Prabhu et al. (2023). Online continual learning without the storage constraint. ArXiv.
> >
> > [2] Kumar et al. (2023). Continual learning as computationally constrained reinforcement learning. ArXiv.
> >
> > [3] Prabhu et al. (2023). Computationally budgeted continual learning: what does matter? CVPR.
> >
> > [4] Elsayed et al. (2024). Addressing loss of plasticity and catastrophic forgetting in continual learning. ICLR.
> >
> > [5] Kirkpatrick et al. (2017). Overcoming catastrophic forgetting in neural networks. PNAS.

---

> > > ### Comment · Reviewer_tK5u · 2024-11-27
> > >
> > > Thank you for the additional experiments, though some questions/comments remain:
> > > 1. While I thank the authors for citing a precedent for this experiment, why not experiment in the more "standard" continual learning setup for measuring catastrophic forgetting reduction? By far the most common setting at this point is the so-called "split-[X]" benchmark, where the classes of a dataset (e.g. CIFAR 10/100, ImageNet) are split into T tasks, which are then learned sequentially, in both task-incremental and class-incremental learning settings. I realize it's pretty late in the discussion period, so I'm not necessarily requiring such an experiment, but generally speaking, such an experiment would be considered more "mainstream" and thus be easier to grasp by others in the field, who are more used to considering only the catastrophic forgetting problem.
> > > 2. Why choose to plot previous task test accuracy in Fig 15, and not something more standard like average accuracy, or per-task accuracy? At first glance, the second row plots seem to give deep Fourier features the advantage, but I believe this is actually misleading. Previous task accuracy is highly dependent on the accuracy the model attained when it was the current task ("giving more room to fall"), so model's with deep Fourier features benefit from a higher starting point compared to the other methods due to its superior plasticity, which can be seen by comparing each second row plot with the top row plots. As such, I don't believe that this figure is actually showing forward transfer as the authors claiming, so much as more evidence of better plasticity. What I'd really be interested to see in this case is what the Task 1 accuracy looks like as the task number increases.
> > > 3. EWC is a classic continual learning method at this point, but also one that has been shown repeatedly to not do very well, particularly in class-incremental learning settings or long task sequences, where it can often fail completely. In future version of this draft, I would encourage exploring other methods as well.
> > > 4. In future versions of this paper, I would encourage including such an experiment in the main body, if space allows, as opposed to the Appendix.
> > >
> > > I thank the authors for shedding more light on their thoughts on focusing on plasticity only. On the other hand, I still hold elements of my original position:
> > > * Even with the relative cheapness of storage, there are still settings where data storage is not a viable option. Examples include applications with sensitive/private data, mobile applications deployed widely, or very long task sequences. Thus catastrophic forgetting without storing data is still a need, because storing data isn't applicable in every scenario.
> > > * Model resets are actually fairly standard practice. Many of the large-scale models deployed by big tech to billions of users are actually reset and retrained on a daily basis. Furthermore, as I said before, I expect progress in self-supervised learning or foundation models to continue to make this cheaper. Even in settings where compute is constrained, as the authors say, the choice can still always be made whether to train the new task from the previous task's weights, or from some reset.

---

> > > > ### Author Response · Authors · 2024-11-28
> > > >
> > > > We appreciate the reviewer's continued engagement.
> > > >
> > > > We have updated Figure 15 to include the accuracy on the first task as the task number increases. These results indicate even more clearly that deep Fourier features are capable of mitigating catastrophic forgetting, compared to both the ReLU and CReLU baseline.
> > > >
> > > > Note that the “split-X” continual learning problems necessarily make each individual task easier by using only a subset of the dataset, and it usually involves only 2 classes per task. The reviewer does note that the problem we consider has precedent in the continual learning literature, but it is also more challenging because it uses the entire dataset for each task. For example, we can readily see that moving from CIFAR10 to CIFAR100 is significantly harder due to the increased number of classes, which a “split-X” problem does not capture.
> > > >
> > > > On previous task accuracy: Note that continual learning often involves both plasticity and stability (“anti-forgetting”, as another reviewer put it). This is why we also included the performance on the current task to help disentangle the benefits of plasticity from the benefits of stability. For example, on label-permuted CIFAR10, we see that all methods do equally well in the current task (and thus, we can conclude that all methods have roughly the same plasticity on this problem). However, there are stark differences in the previous task performance. The updated figure with first task accuracy shows this even more clearly.
> > > >
> > > > On EWC: We chose to use this method precisely because it is a classic method. The fact that EWC works well with deep Fourier features demonstrates that our method is generally effective in continual learning.
> > > >
> > > > About forgetting, plasticity and large models: We agree with the reviewer that storing data is not always desirable or possible. We only point out that the same is true for resetting, such as in computationally-restricted settings, or sustaining performance in an online setting. Also, if data is not stored, then it becomes even more pertinent to not reset the network: the information learned by the network on the previous data cannot be relearned after resetting without storing that previous data. While it is true that the current practice in training large models involves both storing data and resetting, these are limitations that research in catastrophic forgetting and loss of plasticity aims to improve.

---

> > > > ### Author Response · Authors · 2024-12-03
> > > >
> > > > Dear Reviewer tK5u,
> > > >
> > > > The author-reviewer discussion period will be ending today, and we wanted to send this gentle reminder.
> > > >
> > > > We appreciate the reviewer's feedback and their comments on the paper, especially in referring to it as "a breath of fresh air." We hope that our rebuttal and additional results (including forgetting results on the previous and first task) have convinced you of our contribution.
> > > >
> > > > We would like to reiterate a point made in our last comment that highlights the need for studying plasticity as a standalone topic:
> > > >
> > > > >Also, if data is not stored, then it becomes even more pertinent to not reset the network: the information learned by the network on the previous data cannot be relearned after resetting without storing that previous data.
> > > >
> > > > To summarize: there is a need for effective and efficient methods that mitigate loss of plasticity. We contribute to the growing literature on loss of plasticity with a principled perspective, that is more effective and efficient than our baselines (which include full and selective network resets).
> > > >
> > > > We hope the reviewer reconsiders the importance of plasticity as a standalone topic in continual learning, and would love to hear back from the reviewer on whether we have addressed their concerns.

---

### Official Review · Reviewer_TWNH · 2024-10-28

**Soundness:** 3
**Presentation:** 3
**Contribution:** 2
**Rating:** 6
**Confidence:** 4

**Summary:**

This manuscript targets continual learning and the phenomenon of loss of plasticity. The authors propose a theoretical analysis to solve this issue, thereby proposing deep Fourier features that employ sin and cos as activation functions.

**Strengths:**

I find that the manuscript is well-written, including motivation, problem statement, and theoretical analysis. The underlying theory is quite solid and sufficiently described with mathematical details in the appendix.

The use of sin and cos looks plausible to me. Indeed, sigmoid activation function is known to exibit vanishing gradient problem. However, when using sin and cos simultaneously, at least one of the two functions would exhibit an alive gradient. In consideration of this issue, the combination of sin and cos would provide much stable behavior compared with the use of a single one.

**Weaknesses:**

Nevertheless, the most critical issue for this manuscript is the small-scale experiments. Nowadays, experiments on MNIST/CIFAR/tiny-ImageNet with ResNet-18 are too weak to validate the proposed methods. They have only small scale sizes, which is quite deviated from the practical scenario. I believe that the ICLR paper should have a technical bar at least for ImageNet experiments. Note that experiments with ImageNet would require a large number of GPUs and training time (2~3 days per training). I found that you mentioned limited GPU sources, which may be inappropriate to conduct ImageNet experiments. Although my comment may be unfair for your experimental environment, conducting extensive validation with sufficient datasets is critical in our research field, so please understand my concern. Readers want credibility.

**Questions:**

See the weaknesses.

---

> ### Author Response · Authors · 2024-11-24
>
> We thank the reviewer for their comments, particularly praising the submission as well-written, with clear motivation, problem statement and theoretical analysis. We address your main concern below,
>
> >Nevertheless, the most critical issue for this manuscript is the small-scale experiments.
>
> We appreciate the sentiment, as it is now common in machine learning papers to have bigger and bigger datasets. One thing that is easier to overlook, though, is the different empirical methodology used in continual learning research. We are not training a neural network once, but many many times across different tasks. Because of that, experiments in continual learning take much longer than those in more traditional supervised learning literature.
>
> In any case, we share the reviewer's concern for scalability and generality of the proposed approach, and this is exactly why we evaluated our approach in all the major datasets used in the papers published in top-tier venues in the loss of plasticity literature, including Nature. These long training processes have been evaluated primarily using tiny-imagenet, CIFAR100 and CIFAR10. For example, CIFAR10 is used in [1], CIFAR10 and MNIST variants are used in [2,4], and more recently tiny-imagenet is used in [3,5]. The large performance differences in our experiments, particularly on tiny-ImageNet and across different wide residual networks, indicate that the scale of our experiments effectively showcases our contribution and validates the theoretical motivation we provided in Sections 3 and 4.
>
> In contrast, there is no published work studying loss of plasticity on imagenet. Taking your estimate of 2 days for training to convergence on a single task, a single run using imagenet and the label noise problem that we considered would involve 20 days of training (due to training to convergence on 10 tasks).
> We have added additional results on another problem setting (Continual Imagenet, which involves binary classification of two imagenet classes, Appendix D.4), as well as additional architectures (Wide Residual Networks, Appendix D.2.)
>
> [1] Lyle et al. (2023). Understanding Plasticity in Neural Networks. ICML.
>
> [2] Dohare et al. (2024). Loss of plasticity in deep continual learning. Nature (632).
>
> [3] Elsayed et al. (2024). Addressing Loss of Plasticity and Catastrophic Forgetting in Continual Learning. ICLR.
>
> [4] Galashov et al. (2024). Non-Stationary Learning of Neural Networks with Automatic Soft Parameter Reset. NeurIPS.
>
> [5] Lee et al. (2024). Slow and Steady Wins the Race: Maintaining Plasticity with Hare and Tortoise Networks. ICML.

---

> > ### Comment · Reviewer_TWNH · 2024-11-26
> > **Thank you**
> >
> > Thank you for the clarification, and now I understand your point. I admit that the authors conducted additional results, such as Continual ImageNet. I also read other reviews, and as they pointed out, though the use of sine and cosine brings interesting property, it would require further solid explanation. I evaluate this manuscript as near borderline, but I raise the score (5->6).

---

### Official Review · Reviewer_TwPR · 2024-11-04

**Soundness:** 4
**Presentation:** 3
**Contribution:** 3
**Rating:** 5
**Confidence:** 4

**Summary:**

This work investigates loss of plasticity in different continual learning scenarios, which hinders networks' ability to adapt to new knowledge over time. The authors introduce Deep Fourier Features as a new method to enhance plasticity, which integrate sine and cosine functions as activation components in each layer of the network. These features concatenate sine and cosine functions in each network layer, allowing for a balance between linearity (supporting plasticity) and nonlinearity (supporting representational power). Through theoretical and empirical results, the paper demonstrates that using Deep Fourier Features can significantly improve continual learning performance across different tasks and datasets, outperforming traditional ReLU-based networks.

**Strengths:**

+ The paper presents a novelty method by integrating Fourier Features into deep networks, offering a new view on balancing linearity and nonlinearity to sustain plasticity.
+ The authors provide a comprehensive theoretical basis for their Deep Fourier Feature, proving that linear function approximations and certain linear networks maintain plasticity.
+ The experiments were conducted on benchmark datasets under various continual learning scenarios.

**Weaknesses:**

+ Although they compare the proposed method with ReLU-based architectures, it could include comparisons with other advanced techniques in continual learning to validate the advantages further.
+ There is limited discussion on the computational and memory costs associated with the model compared to standard activation functions.
+ The authors claim that the proposed method focuses on the loss of plasticity, however, there seems to be something confusing about the experiments on label noise and pixel permutations (especially label noise).
+ The manuscript could benefit from more discussion of the overfitting problem.
+ The manuscript could benefit from more practical insights or guidelines for implementing deep Fourier features.
+ More continual learning methods related to plasticity should be included, like

[1] Architecture matters in continual learning

[2] Revisiting Neural Networks for Continual Learning: An Architectural Perspective

**Questions:**

plz see Weaknesses.

---

> ### Author Response · Authors · 2024-11-24
>
> We appreciate the reviewer’s thoughtful questions regarding our submission. Several suggestions have been used to add clarifying sentences (all changes and new content is in blue, except for minor typos which have been corrected). We address your comments below,
>
> >Although they compare the proposed method with ReLU-based architectures, it could include comparisons with other advanced techniques in continual learning to validate the advantages further.
>
> We consider baselines of all prominent approaches for dealing with loss of plasticity, as described in lines 428-431. Could the reviewer clarify what they mean by “other advanced techniques”?
>
> >There is limited discussion on the computational and memory costs associated with the model compared to standard activation functions.
>
> Thanks for bringing this up, we have added a sentence describing this in Section 4. The resnet that uses our proposed deep Fourier features has approximately half the computation and memory when compared to the vanilla resnet. This is because we use half the output width in each layer due to concatenating sine and cosine.
>
> >The authors claim that the proposed method focuses on the loss of plasticity, however, there seems to be something confusing about the experiments on label noise and pixel permutations (especially label noise).
>
> The performance curves for the label noise experiments demonstrate that the baselines are unable to learn as effectively when trained on noisy labels that are progressively made into clean labels. This indicates loss of plasticity, where the plasticity present at initialization decreases on later tasks. We are happy to adjust the text accordingly to clarify any potential confusion, but it is unclear to us what is confusing right now.
>
> >The manuscript could benefit from more discussion of the overfitting problem.
>
> If the reviewer refers to the fact that Fourier features can overfit, then this is a well-documented spectral bias, in which Fourier features can learn high-frequency features that are more prone to overfitting. We have added clarification on this point in Section 5.2
>
> >The manuscript could benefit from more practical insights or guidelines for implementing deep Fourier features.
>
> We have added pseudocode in Appendix A.4. We would like to highlight that deep Fourier features are straightforward to implement. It requires i) concatenating the sine and cosine activation functions at every layer, and ii) that the layers have half their output width to accommodate the concatenation. We also added a sentence describing the implementation to Section 4.2.
>
> >More continual learning methods related to plasticity should be included, like [1,2]
>
> We appreciate the references, but neither of these papers deal with the loss of plasticity phenomenon. Our experiments already include representatives of all prominent baselines: including regularization, weight-reinitialization and activation functions. The references provided include completely different architectures, which would not provide any meaningful comparison as a baseline relative to the results presented in our paper.
>
> However, we have added additional experiments using more architectures, please see “Wide Residual Networks:” in shared reply.

---

> > ### Author Response · Authors · 2024-11-27
> > **Gentle reminder**
> >
> > Dear Reviewer,
> >
> > We would like to thank you for the time spent reviewing our submission.
> >
> > The main discussion phase will be ending soon, and we wanted to send this gentle reminder. We have done our best to answer the comments you raised, as well as incorporate your suggestions. We would love to hear back from the reviewer and whether we have addressed their concerns.

---

> > ### Comment · Reviewer_TwPR · 2024-11-28
> >
> > Thanks to the authors for their responses. However my concerns were not well addressed. Therefore, I keep my score.
> > For example, in response to my concerns, I don't seem to be getting useful information from the author's rebuttal.

---

### Official Review · Reviewer_GT8H · 2024-11-04

**Soundness:** 4
**Presentation:** 3
**Contribution:** 3
**Rating:** 8
**Confidence:** 4

**Summary:**

The paper studies the problem of plasticity loss in deep neural network under continual learning. The authors suggest that the loss of plasticity or trainability is caused by the nonlinearity in activation functions.  The authors prove that linear layer and a diagonal linear network dont suffer from a loss of plasticity, showing empirically that deep linear network also dont suffer from that   The authors then suggest that a perfect mix of linearity and nonlinearity  is a potential solution to the plasticity loss while maintain the expressiveness of deep neural network.
The authors then suggest deep Fourier features where sin and cosine activation functions are combined. The authors showed in the experiments that such activation function maintain trainability but needs regularization to prevent overfitting.

**Strengths:**

The paper is well written.

A discussion and analysis of an important problem.

Showing that linear network don't suffer plasticity loss and a mix between linear and non linear can retrain trainability.

Theoretical evidence and empirical evidence support the paper contribution.

**Weaknesses:**

While I enjoyed reading the paper and its flow, the presentation can still be improved substantially especially in the figures and equations.
For example, equations are not numbered which makes it hard to refer back to specific equation. Figures are not well explained, and colours are really hard to distinguish among different baselines.

While a new activation function is introduced, a proper analysis of its behaviour and the performance of the network should be demonstrate here we only see results on continual learning and later figure out that the results were with a regularization.

It is not clear which metrics are reported, e.g., test accuracy measured by what? performance on the latest task or all tasks?
Forgetting is never discussed.

Only one network is studied, what about other architectures?

**Questions:**

Why loss of trainability is used, does it differ from the loss of plasticity defined in https://www.nature.com/articles/s41586-024-07711-7 and if not why using different terms?

What is the performance compared to ReLU or LeakyReLU activations on offline setting (training on full dataset with no continual learning)?


Ln 076-079 are not quite clear

Could you define unit sign entropy earlier, or make it clear when using it in the intro.

ln 107 is unclear.

What is the suboptimality gap in line 144

In ln 305, what if \alpha was trainable parameter?

Could you add continual backpropagation as a baseline https://www.nature.com/articles/s41586-024-07711-7

Please add a section to describe the metrics used.

It is unclear to me why the setup of diminishing noise is considered as the first experiment.

Does the trainability still persist on longer tasks, e.g., the ImageNet sequence of   https://www.nature.com/articles/s41586-024-07711-7

I am welling to increase my score upon addressing my questions, thanks!

---

> ### Author Response · Authors · 2024-11-24
>
> We are pleased that the reviewer enjoyed reading our submission and appreciate the suggestions for improvement. The updated submission incorporates several of these suggestions, including experiments in the single task setting, additional architectures and results on continual imagenet (all changes and new content is in blue, except for minor typos which have been corrected).  We address individual comments below,
>
> >While a new activation function is introduced, a proper analysis of its behaviour and the performance of the network should be demonstrate here we only see results on continual learning and later figure out that the results were with a regularization.
>
> The sensitivity analysis in Section 5.2 explicitly investigates the use of deep Fourier features with and without regularization, which allows us to fully characterize the impact of each component. We find that unregularized deep Fourier features can sustain trainability, but can lack comparatively in generalization. While it is well-documented that Fourier features overfit without regularization due to their spectral bias, we find the surprising result that the increased trainability of deep Fourier features enables training with very large regularization strengths. This combination of increased trainability and high regularization strength leads to much stronger generalization, as demonstrated by Figures 5 and 6.
>
> >It is not clear which metrics are reported, e.g., test accuracy measured by 3what? performance on the latest task or all tasks? Forgetting is never discussed.
>
> Our evaluation is consistent with the problem setting we describe in Section 2 (lines 124-126): The network is evaluated on data from the latest task as is commonly done in the loss of plasticity literature.
>
> Please refer to “On forgetting:” in the shared reply.
>
> In addition, note that the experiment setting that we use isolates the loss of plasticity phenomenon and does not involve catastrophic forgetting. For example, each task in class incremental learning involves all previous classes, as in [1]. The label noise experiment also does not involve catastrophic forgetting, in fact the goal is to forget the initial noisy labels as they become progressively clean [2].
>
> [1] Dohare et al. (2024). Loss of plasticity in deep continual learning. Nature (632).
>
> [2] Lee et al. (2024). Slow and Steady Wins the Race: Maintaining Plasticity with Hare and Tortoise Networks. ICML.
>
> >Only one network is studied, what about other architectures?
>
> Thanks for bringing this up. For clarity, and to avoid too much clutter in the results, we had decided to present, in the main paper, results mostly for only one large architecture configuration. Nevertheless, notice that our results in Sections 3 and 4, and in the Appendix, investigate MLPs of varying widths and depths.
> In any case, we have now added an additional experiment studying additional large architectures, particularly Wide Residual Networks [3], to study the effect of width on Deep Fourier Features.
>
> [3] Zagoruyko and Komodakis. (2016). Wide Residual Networks. ArXiv
>
> >Why loss of trainability is used, does it differ from the loss of plasticity defined in https://www.nature.com/articles/s41586-024-07711-7 and if not why using different terms?
>
> Loss of plasticity is used ambiguously in the literature, and can refer to loss of trainability or to loss of generalization. We use the term loss of trainability to eliminate this ambiguity. Our theoretical focus is on trainability using an optimization perspective. In contrast, a theoretical understanding for loss of generalization would require understanding the mechanisms by which neural networks generalize. Understanding generalization of neural networks is an active research area, without guiding theory even outside of continual learning. We have added a sentence in Section 2 clarifying this.
> Note, however, that our experiments evaluate both trainability and generalization.
>
> >What is the performance compared to ReLU or LeakyReLU activations on offline setting (training on full dataset with no continual learning)?
>
> We have added an additional figure to Appendix D.3. that compares performance on the full tiny-ImageNet dataset without any nonstationarity, using different activation functions. We see that deep Fourier features are highly trainable, and can generalize well with regularization.
> Note that the performance curve in Figure 5 on the first task is similar to learning in a non-continual setting (albeit, with some noisy labels making the problem harder). In this setting, we see that early performance across all methods is similar.

---

> > ### Author Response · Authors · 2024-11-24
> >
> > >Could you add continual backpropagation as a baseline
> >
> > Note that continual backprop is a weight-reinitialization method and dormant neuron recycling (ReDO) is also a weight reinitialization method. Moreover, it was found in the ReDO paper that ReDO outperforms or performs similarly compared to continual backprop. This is why we initially  used ReDO as a representative weight reinitialization method. In any case, we have now added some experiments to corroborate the fact that ReDO performs similarly to continual backprop (see Appendix D.2 and Appendix D.5.)
> >
> > >It is unclear to me why the setup of diminishing noise is considered as the first experiment.
> >
> > The diminishing label noise experiment is a recently studied problem, motivated from the related problem of warm-staring [4]. We have added a clarifying sentence to motivate this.
> >
> > [4] Lee et al. (2024). Slow and Steady Wins the Race: Maintaining Plasticity with Hare and Tortoise Networks. ICML.
> >
> > >Does the trainability still persist on longer tasks, e.g., the ImageNet sequence of https://www.nature.com/articles/s41586-024-07711-7
> >
> > It is unclear if the reviewer means “longer tasks” (more iterations per task) or “more tasks.” The experiment in the linked paper (“continual imagenet”) does have many tasks, but it is actually much easier than the problems that we consider. In particular, it does not require many iterations to learn the task because each task is a binary classification problem. We have added additional experiments showing sustained performance on continual imagenet.
> >
> > >While I enjoyed reading the paper and its flow, the presentation can still be improved substantially especially in the figures and equations. For example, equations are not numbered which makes it hard to refer back to specific equation. Figures are not well explained, and colours are really hard to distinguish among different baselines.
> >
> > Thank you, we have added equation numbers and adjusted some of the colors to improve visual clarity among different baselines. Could the reviewer clarify which figures are not well-explained? These are easy fixes for the final version of the paper, and we will appreciate any additional feedback.
> >
> > >Ln 076-079 are not quite clear
> >
> > We agree with the reviewer: there is a lot going on in that sentence. We separated it into multiple sentences for additional clarity:
> >
> > “The plasticity of deep linear networks is surprising compared to the loss of plasticity of deep nonlinear networks. This finding suggests that loss of plasticity is not necessarily caused by the nonlinear learning dynamics, but a combination of nonlinear learning dynamics and nonlinear representations.”
> >
> > >Could you define unit sign entropy earlier, or make it clear when using it in the intro.
> >
> > We have added additional explanation to line 89 to provide more intuition on the unit sign entropy before defining it formally in Section 4.
> >
> > >ln 107 is unclear.
> >
> > This statement alludes to our results in Section 5.2, where we show that networks with deep Fourier features can be trained with much larger regularization strengths to improve generalization performance in continual learning. We have added these details to clarify the statement.
> >
> > >”What is the suboptimality gap in line 144”
> >
> > The suboptimality gap in optimization is the difference between the solution found by the optimizer and the optimum. Specifically, it refers to the term $J_\tau (\theta^{(τ T )}) - J_\tau (\theta^\star)$. We have added an additional explanation to the submission to make this explicit.
> >
> > >In ln 305, what if \alpha was trainable parameter?
> >
> > This is a great suggestion, and one that we briefly explored in earlier work. However, as our results in Figure 2 indicate, avoiding loss of trainability with a larger value of $\alpha$ still leads to the problem of slow learning with overly linear representations. Gradient-based training aims to minimize the loss, and large $\alpha$ values — that avoid loss of trainability — are slow to train. Thus, gradient-based (steepest-descent) training of $\alpha$ would lead to small $\alpha$ and loss of trainability (rather than larger values of $\alpha$ that sustain trainability). This is why we emphasize in line 309 that α-linearization is only to gain insights from empirical investigation, and it is not a solution to loss of trainability.
> >
> > >Please add a section to describe the metrics used.
> >
> > We have added an additional description of the metrics to Appendix C.

---

> > > ### Comment · Reviewer_GT8H · 2024-11-26
> > > **Thank you for your feedback**
> > >
> > > I thank the authors for their response, they have addressed my concerns. I will increase my score.
> > > One question, when going to a larger architecture, a wide ResNet is chosen. What about transformer? or even only one transformer block, is the loss of trainability shown their? Would the proposed activation work their?

---

> > > > ### Comment · Reviewer_GT8H · 2024-12-02
> > > > **Could you please address my question**
> > > >
> > > > I am still interested in understanding the link between the proposed features and the transformer architecture, and loss of trainability.

---

> > > > > ### Author Response · Authors · 2024-12-02
> > > > >
> > > > > Thanks for the great question. We chose the Wide Residual Network because it can be configured uniformly at different width scales (rather than pre-configured Vision Transformers), without quadratic scaling in the number of patches due to self-attention. Loss of plasticity has been demonstrated when using a vision transformer [1], and it has also been shown that transformers are less trainable than ResNet, despite achieving higher generalization performance [2].
> > > > >
> > > > > It is straightforward to switch the activation functions in the MLP of the vision transformer, from GeLU to Fourier, which can improve continual trainability. Our primary motivation, maintaining approximate linearity, would not hold in this architecture because of self-attention. However, state-space models or linearized attention may be particularly well-suited to deep Fourier features.
> > > > >
> > > > > [1] Lee et al. (2024). Slow and Steady Wins the Race: Maintaining Plasticity with Hare and Tortoise Networks. ICML.
> > > > >
> > > > > [2] Lewandowski et al. (2024). Learning Continually by Spectral Regularization. ArXiv.

---

### Official Review · Reviewer_rXxi · 2024-11-08

**Soundness:** 3
**Presentation:** 4
**Contribution:** 2
**Rating:** 6
**Confidence:** 3

**Summary:**

Continuous learning involves a model learning from changing, non-stationary distributions. The model should avoid forgetting previously learned information while adapting to new incoming data. The paper first demonstrates that linear networks adapt more easily to new data than non-linear models. However, non-linearity is crucial for capturing features and achieving good performance in supervised settings. To balance linearity and non-linearity, the paper proposes using an activation function that combines cosine and sine functions, which is called deep fourier features.

**Strengths:**

The proposed activation funtion may replace the ReLU and serve as the comman practice when building continuous learning model.

**Weaknesses:**

1. While the authors establish that linear networks can more readily adapt to new data (Sec 3.1), they do not examine whether these networks are resistant to catastrophic forgetting. Without measuring anti-forgetting properties, it’s unclear if the proposed approach truly meets both requirements of continuous learning (adaptability and retention).

2. The proposed activation function, $Fourier(z)=[sin(z),cos(z)]$, lacks clear implementation details, which may hinder practical use and replication. For instance, it’s ambiguous how the function applies to each unit in a layer and whether each pre-activation value is mapped to both sine and cosine or alternates between the two.

3. While using sine and cosine for the activation function is interesting, the rationale behind this choice is not fully explained. Periodic linear functions might also be a candidate for balancing linearity and non-linearity, so further justification or comparative analysis would strengthen the argument for cosine and sine.

**Questions:**

1. The theory in Section 3.1 suggests that linear networks can adapt to new data, but does not address resistance to forgetting. Could the authors provide empirical results to assess the anti-forgetting capability of their approach? This would help clarify whether the method supports both critical aspects of continuous learning.

2. Can the authors explain how the Fourier activation function, defined as $Fourier(z)=[sin(z),cos(z)]$, is applied in practice? For instance, if a pre-activation vector is given as $[z_1, z_2, ..., z_n]$, how would each unit in the layer be processed? Should each pre-activation value use both sine and cosine, or is there a selection mechanism between the two? Pseudo-code or a mathematical expression would be highly helpful.

3. Why did the authors choose cosine and sine for the activation function over other periodic functions or piecewise linear functions within defined intervals (e.g., periodic $y = x$ replacing $ y = sin(x)$, periodic $y = 1-|x|$ replacing $ y = cos(x)$)? A justification of this choice would provide greater insight into the effectiveness of these functions for balancing linearity and non-linearity in continuous learning.

Addressing these questions could strengthen the paper’s contribution and may support a higher rating score.

**Details Of Ethics Concerns:**

No Ethics Concerns

---

> ### Author Response · Authors · 2024-11-24
>
> We thank the reviewer for their thoughtful comments and questions regarding our submission. We have added several clarification sentences to the submission in light of your feedback (changes and new content is in blue). Below, we address your individual questions.
>
>
> >the authors [...] do not examine whether these networks are resistant to catastrophic forgetting.
>
> Please see “On forgetting:” in the shared reply.
>
> >The proposed activation function lacks clear implementation details [...] it’s ambiguous how the function applies to each unit in a layer and whether each pre-activation value is mapped to both sine and cosine or alternates between the two.
>
> Thanks for bringing this up, we did not realize our presentation was ambiguous. We have added a clarifying sentence in Section 4.2 and pseudocode . Shortly, when talking about the implementation: In Figure 1 and line 344, we define the Fourier activation function as mapping each pre-activation to both sine and cosine. Having the pre-activation mapping to both sine and cosine is critical for approximate linearity discussed in the paragraph on line 399: when one of the activation functions is in a nonlinear region, the other activation function is in a linear region.
>
>
> >Periodic linear functions might also be a candidate for balancing linearity and non-linearity, so further justification or comparative analysis would strengthen the argument for cosine and sine.
>
> On triangle wave: The reviewer’s suggestion of using linear periodic functions (triangle wave) is interesting. The main downside of triangle waves is that they are unstable to train because of exponentially many non-differentiable points. While relu has a non-differentiable point at the origin, the origin admits a reasonable choice of 0 for the subdifferential. For a triangle wave, however, there are exponentially many non-differentiable points away from zero which do not have a reasonable choice of subdifferential. For example, if the triangle wave has a peak at x = 1, then the subdifferential can be any number between [-1,1]. This leads to a lack of stability during training, which explains why they have not been used before.
>
> In contrast, sine and cosine are smooth and have the property of closure under differentiation (the gradients are also sines and cosines). They also have a rich existing connection with previous neural network architectures through shallow Fourier features.
>
>
> >The theory in Section 3.1 suggests that linear networks can adapt to new data, but does not address resistance to forgetting.
>
> Please see “On forgetting:” in the shared reply.
>
> In addition to that, note that the experiment setting that we use does not involve catastrophic forgetting. This experiment setting is commonly used in loss of plasticity literature, see class-incremental setting in [1] and the label noise experiment in [2]. For example, the class incremental tasks involve all previous classes, as in [1]. The label noise experiment also does not involve catastrophic forgetting, in fact the goal is to forget the initial noisy labels as they become progressively clean.
>
> [1] Dohare et al. (2024). Loss of plasticity in deep continual learning. Nature (632).
>
> [2] Lee et al. (2024). Slow and Steady Wins the Race: Maintaining Plasticity with Hare and Tortoise Networks. ICML.
>
>
> >Can the authors explain how the Fourier activation function, is applied in practice? [...] Pseudo-code or a mathematical expression would be highly helpful.
>
> We have added pseudocode to Appendix A.4. The expression provided, $\text{Fourier}(z) = [\sin(z), \cos(z)]$, summarizes the implementation which involves concatenating the element-wise sine and element-wise cosine of the pre-activation. Specifically, given a $d$ dimensional pre-activaiton, $z \in \mathbb{R}^d$, the Fourier activation is a mapping to the higher dimensional space: $\text{Fourier}: \mathbb{R}^d \rightarrow [-1,1]^{2d}$. This is why we comment in Section 4 that our baseline actually has approximately half the parameters, every layer has half the width to accommodate the concatenation.
>
>
> >Why did the authors choose cosine and sine for the activation function?
>
>
> Thanks for bringing this up. We describe some of the rationale in Section 4 (line 361), with sine and cosine as natural choices that have been explored in other contexts (e.g., shallow Fourier features). To elaborate: A single periodic function does not provide approximate linearity in the parameters. Other concatenated activation functions involving piecewise linearity, like concatenated ReLU, provide linearity but are not adaptive due to a low unit sign entropy. In contrast, a pair of periodic activation functions achieves both i) approximate linearity and ii) high unit entropy (with many units active for any range of pre-activations, due to periodicity). We have added a few sentences to Section 4.2 before proposition 1 for additional clarity.

---

> ### Author Response · Authors · 2024-11-27
> **Gentle reminder + anti-forgetting results**
>
> Dear Reviewer,
>
> We would like to thank you for the time spent reviewing our submission.
>
> The main discussion phase will be ending soon, and we wanted to send this gentle reminder. We have done our best to answer the comments you raised, as well as incorporate your suggestions. We would love to hear back from the reviewer and whether we have addressed their concerns.
>
> Note: We now have updated the submission with several new results demonstrating deep Fourier features can also provide benefits in avoiding catastrophic forgetting (see Appendix D.7). Our experiment uses the setting described by [4], in which the labels for a dataset are permuted at the beginning of each task, and learning is online, meaning that each data point is seen exactly once. Moreover, and unlike the experiments in [4], we use more large-scale datasets (+CIFAR100/tiny-ImageNet) and train a ResNet end-to-end rather than just the last two layers. A good continual learning algorithm on this problem should both i) improve as more tasks are seen and ii) maintain performance on previously seen tasks. For all methods considered (ReLU, CReLU and deep Fourier features), we use elastic weight consolidation to regularize towards the parameters of the previous task [5]. Our results demonstrate that deep Fourier features can sustain high accuracy on the current and previous task, and that deep Fourier features are robust to the hyperparameter strength across online label-permuted tiny-ImageNet, CIFAR10 and CIFAR100.

---

> > ### Comment · Reviewer_rXxi · 2024-11-27
> >
> > Thanks for the additional experiments. I think these address my concerns. So I will raise my score from 5 to 6.

---

### Author Response · Authors · 2024-11-24

We appreciate all of the reviewers' comments and suggestions. The reviewers have a generally positive assessment, in they refer to the submission as “a breath of fresh air” [tK5u], and “theoretical evidence and empirical evidence support the paper contribution” [GT8H].

Our goal in this rebuttal is to address the reviewers comments thoroughly. We have updated the submission to include several new results:

* Wide Residual Networks: Using different width scales, we found that deep Fourier features are effective at continual learning on tiny-ImageNet at both small and large widths, despite using approximately half the parameters of the other baselines (see Appendix D.2).

* Additional baselines: in our updated results, we also include a full network reset as a pseudo-oracle that is given privileged access to the task boundary information. We also included continual backprop, with results showing that our choice of ReDO as a representative weight reinitialization method is justified (See Appendix D.2 and Appendix D.5).

* Additional continual learning problem: We show that deep Fourier features can sustain their performance over 500 tasks on Continual ImageNet from [6] (See Appendix D.4).

* Other new results include additional ancillary results for the case-studies in Sections 3 and 4 (Appendix D.1), and single task (non-continual) performance (Appendix D.3)

On forgetting: A couple of reviewers brought up the problem of catastrophic forgetting. We agree this is a very interesting problem when thinking about continual learning. We want systems that can learn forever, ideally forgetting as little as possible. Nevertheless, it is important to realize that a continual learning system should constantly adapt because it can’t learn everything about the world, an idea sometimes referred to as the big world hypothesis [1, 2]. In that case, we then may want our systems to forget so that it can use its capacity to learn from the most recent data. A neural network with limited capacity can only succeed on future tasks by forgetting previous ones. This is not a foreign idea in the field. In fact, it has been shown that using data from previous distributions is an effective way of mitigating forgetting [3]. In that sense, the tasks in our class incremental experiments account for this to some extent. We will further emphasize this in our manuscript, also adding a discussion on catastrophic forgetting in Section 1.

That being said, we avoided talking about forgetting in most of our paper because our paper is explicitly about plastic learning. Dealing with loss of plasticity is an important subproblem of continual learning, with many examples of papers focusing on this problem alone [4, 5], including a paper recently published in Nature that explicitly “focus on plasticity rather than on forgetting” [6]. We provide several insights into loss of plasticity, particularly the effectiveness of linearity in sustaining plasticity and using deep Fourier features for approximate linearity. In fact, linearity has recently been shown to be resistant to forgetting under certain task-ordering assumptions [7]. We also added this discussion in Section 1.

[1] Javed and Sutton. (2024). The big world hypothesis and its ramifications for artificial intelligence. RLC Finding the Frame Workshop.

[2] Kumar et al. (2024). The Need for a Big World Simulator: A Scientific Challenge for Continual Learning. ArXiv.

[3] Prabhu et al. (2020). Gdumb: A simple approach that questions our progress in continual learning. ECCV.

[4] Lyle et al. (2023). Understanding Plasticity in Neural Networks. ICML.

[5] Lee et al. (2024). Slow and Steady Wins the Race: Maintaining Plasticity with Hare and Tortoise Networks. ICML.

[6] Dohare et al. (2024). Loss of plasticity in deep continual learning. Nature (632).

[7] Evron et al. (2022). How catastrophic can catastrophic forgetting be in linear regression? COLT.

---

### Meta-Review · Area_Chair_KZps · 2024-12-21

**Metareview:**

This work studies the reasons behind the loss of plasticity in continual learning, where plasticity refers to a model's ability to acquire new knowledge (after being trained on a previous set of tasks). The authors begin by studying plasticity in linear models and demonstrate that linear models do not experience a loss of plasticity. Building on this, the authors propose linearizing the activation functions in neural networks. This approach is studied in detail, culminating in a solution that leverages the properties of Fourier features, using sine and cosine functions as activation functions. The proposed method is evaluated on a ResNet18 model using a subset of the ImageNet dataset, among other tests. The reviewers were positive about the work overall following the author-reviewer discussion period. The AC concurs with their assessment, recognizing that this work provides valuable insights into the mechanisms of acquiring knowledge in a continual learning setting and hence recommends `acceptance.` Congratulations on this achievement!

**Additional Comments On Reviewer Discussion:**

The paper was reviewed by five experts in continual learning, initially receiving four negative scores and one positive score. During the discussion, the authors addressed the reviewers' concerns, including catastrophic forgetting in the context of this work, providing new experiments and results, along with engaging in constructive dialogues to clarify their contributions.  Following the author-reviewer discussion period, four reviewers raised their scores to positive.  The AC concurs with the reviewers' assessments and is pleased to recommend `acceptance` of the paper.

---

### Decision · Program_Chairs · 2025-01-22

Accept (Poster)